# Prediction of Cyclosporin-Mediated Drug Interaction Using Physiologically Based Pharmacokinetic Model Characterizing Interplay of Drug Transporters and Enzymes

**DOI:** 10.3390/ijms21197023

**Published:** 2020-09-24

**Authors:** Yiting Yang, Ping Li, Zexin Zhang, Zhongjian Wang, Li Liu, Xiaodong Liu

**Affiliations:** 1Center of Drug Metabolism and Pharmacokinetics, China Pharmaceutical University, Nanjing 210009, China; 1821010209@stu.cpu.edu.cn (Y.Y.); lp15850657822@163.com (P.L.); 1821010211@stu.cpu.edu.cn (Z.Z.); 2Jiangsu Key Laboratory of New Drug Research and Clinical Pharmacy, Xuzhou Medical University, Xuzhou 221004, China; cpu_wzhj@163.com

**Keywords:** drug–drug interaction, the interplay of transporters and enzymes, physiologically based pharmacokinetic model, drug transporters, pharmacokinetics

## Abstract

Uptake transporter organic anion transporting polypeptides (OATPs), efflux transporters (P-gp, BCRP and MRP2) and cytochrome P450 enzymes (CYP450s) are widely expressed in the liver, intestine or kidney. They coordinately work to control drug disposition, termed as “interplay of transporters and enzymes”. Cyclosporine A (CsA) is an inhibitor of OATPs, P-gp, MRP2, BCRP and CYP3As. Drug–drug interaction (DDI) of CsA with victim drugs occurs via disordering interplay of transporters and enzymes. We aimed to establish a whole-body physiologically-based pharmacokinetic (PBPK) model which predicts disposition of CsA and nine victim drugs including atorvastatin, cerivastatin, pravastatin, rosuvastatin, fluvastatin, simvastatin, lovastatin, repaglinide and bosentan, as well as drug–drug interactions (DDIs) of CsA with nine victim drugs to investigate the integrated effect of enzymes and transporters in liver, intestinal and kidney on drug disposition. Predictions were compared with observations. Most of the predictions were within 0.5–2.0 folds of observations. Atorvastatin was represented to investigate individual contributions of transporters and CYP3As to atorvastatin disposition and their integrated effect. The contributions to atorvastatin disposition were hepatic OATPs >> hepatic CYP3A > intestinal CYP3As ≈ efflux transporters (P-gp/BCRP/MRP2). The results got the conclusion that the developed PBPK model characterizing the interplay of enzymes and transporters was successfully applied to predict the pharmacokinetics of 10 OATP substrates and DDIs of CsA with 9 victim drugs.

## 1. Introduction

Drug influx transporters, efflux transporters and metabolic enzymes such as cytochrome P450 enzymes (CYP450s) and UDP-glucuronosyltransferases (UGTs) are widely expressed in the liver, intestine or kidney. They coordinately work to control the disposition of the drug in vivo, called as “interplay of transporters and enzymes” [1,2,3,4]. CYP3A4, the main CYP450s, accounts for 30–40% of hepatic CYP450s and mediates 60–70% of drug metabolism [5]. CYP3A4 is also abundant in the intestine, thus being implicated in the first-pass effect of orally administrated drugs. Uptake transporters organic anion transporting polypeptides (OATPs) such as OATP1B1, OATP1B3 and OATP2B1, are mainly expressed at the basolateral surface of hepatocytes and mediate hepatic uptake of their substrates from portal blood, becoming the limited step of hepatic clearance of some drugs such as statins [6]. Efflux transporters mainly include P-glycoprotein (P-gp), multidrug resistance-associated proteins (MRPs) and breast cancer resistance protein (BCRP). They are expressed in the brush border membrane of enterocytes, bile canalicular membrane of hepatocytes and apical membranes of proximal tubules, exporting their substrates out of cells to the intestinal lumen, bile and urine, respectively [7]. Some drugs are often substrates of these transporters and CYP450s. For example, atorvastatin is a substrate of P-gp, BCRP, MRP2, OATPs and CYP3A4, indicating that the disposition of atorvastatin should be attributed to the integrated effects of P-gp, BCRP, MRP2, OATPs and CYP3A4 [7,8,9,10] in the intestine and liver. Another example is cyclosporin A (CsA). CsA is a substrate of P-gp, OATPs and CYP3A4. Moreover, CsA is also a strong inhibitor of P-gp, BCRP, MRP2, OATPs and CYP3A4 [11,12,13]. Accumulating evidences have demonstrated that coadministration of CsA may lead potential drug–drug interactions (DDIs) with victim drug such as atorvastatin [14,15,16], cerivastatin [17], pravastatin [18,19], rosuvastatin [20], fluvastatin [21], simvastatin [22], lovastatin [18], repaglinide [23] and bosentan [24] via inhibiting these transporters or CYP3A4. Hepatic uptake of these drugs is mainly mediated by hepatic OATPs, which leads to high K_p_ values of drugs, then metabolized by CYP450s (main CYP3A4) and biliary or renal excretion via a P-gp, BCRP or MRP2-dependent mechanism. All these processes may be remarkably inhibited by CsA, indicating that CsA-mediated DDIs should be attributed to integrated effects of inhibiting transporters and CYP3A4 in the intestine, liver and kidney. Physiologically based pharmacokinetic (PBPK) model has been widely used to quantitatively predict the pharmacokinetics of drugs and the magnitude of DDIs [25]. Moreover, PBPK also illustrates individual contributions of transporters and enzymes to drug disposition as well as their integrated effects [26].

There have been researches about the DDI between statin and CsA with the PBPK model [27,28], but few have integrated simultaneously of the liver, intestinal and kidney on drug disposal. The highlight of our work is to investigate the integrated effect of enzymes and transporters in the liver, intestinal and kidney on drug disposition using a whole-body PBPK model characterizing the interplay of the enzyme-transporter in the liver, intestine and kidney. The developed PBPK first was used to predict the disposition of CsA and nine victim drugs (OATP substrates) including atorvastatin, cerivastatin, pravastatin, rosuvastatin, fluvastatin, simvastatin, lovastatin, repaglinide and bosentan, as well as DDIs of CsA with nine victim drugs. The predictions were compared with clinical reports. Atorvastatin, the substrate of uptake transporter (OATP), efflux transporter (BCRP, P-gp and MRP) and enzyme CYP3A, was represented to further investigate individual contributions of transporters and CYP3A4 to the disposition of drugs and to document sensitivity analysis.

## 2. Results

### 2.1. Collection of Drug Parameters Used for PBPK Prediction

Physiological parameters of humans such as organ volume, blood flow rate, metabolic and transport parameters of CsA and victim drugs were cited from publications PubMed. Ratios of drug concentration in tissue-to-plasma were cited from previous reports or estimated using the previously reported concentration in tissues and plasma or calculated using the previous method [29] based on tissue composition and physicochemical parameters of drugs. These parameters were list in Table 1, Table 2, Table 3 and Table 4. Based on the DDIs for criterions, nine victim drugs were selected including atorvastatin (substrate of P-gp, BCRP, MRP2, OATPs and CYP3A4) [8], cerivastatin (OATPs, P-gp, CYP3A4 and CYP2C8) [30,31], pravastatin (OATP and MRP2) [9,32], rosuvastatin (P-gp, BCRP and MRP2, OATPs) [8], fluvastatin (substrate of OATPs, P-gp, BCRP, MRP2, CYP2C9 and CYP3A4) [8,33], simvastatin (substrate of P-gp, OATPs and CYP3A4) [31,32], lovastatin (substrate of MRP2, OATPs and CYP3A4) [34], repaglinide (P-gp, BCRP, OATPs, CYP2C8 and CYP3A4) [35] and bosentan (MRP2, OATPs, CYP3A4 and CYP2C9) [36,37]. The CsA-mediated DDI data came from fifteen clinic reports as showed in follow-up sections.

### 2.2. Quantitative Prediction of CsA and Victim Drugs Using the Developed PBPK Model

Pharmacokinetic profiles (Figure 1) and main pharmacokinetic parameters (Table 5) of perpetrator CsA and nine victim drugs (atorvastatin, cerivastatin, pravastatin, rosuvastatin, fluvastatin, simvastatin, lovastatin, repaglinide and bosentan) were predicted using the developed PBPK model and parameters listed in Table 1, Table 2, Table 3 and Table 4. Model performance was assessed using absolute average fold error (AAFE) around each data point, as well as the ratio between observed and predicted peak concentration (C_max_) and area under the curve (AUC). The results showed that 62% predicted concentrations (24/39) and 96% predicted exposure parameters (75/78) fell within 0.5–2.0-fold of observations, inferring that the developed PBPK model was suitable for illustrating pharmacokinetic behaviors of CsA and the tested victim drugs.

### 2.3. Quantitative DDIs of CsA with Victim Drugs

CsA is a strong inhibitor of P-gp, BCRP, MRP2, OATP1B1 and CYP3A4, whose K_i_ values were reported to be 0.895 μM [91] 0.28 μM [55], 4.1 μM [27], 0.014 [27] μM and 2 μM [27], respectively. The DDIs of CsA co-administrated with nine victim drugs were predicted using the developed PBPK and compared with clinic observations. The DDIs were indexed as the ratio of AUC (AUCR) or C_max_ (C_max_ R) for victim drugs with CsA to without CsA. The dosage of CsA was set to be the highest dosage reported in clinic reports. The Pharmacokinetic profiles for coadministration of perpetrator CsA and victim drugs (atorvastatin, cerivastatin, pravastatin, rosuvastatin, fluvastatin, simvastatin, lovastatin, repaglinide and bosentan) were showed in Figure 2, Appendix A and Table 6. The detailed interpretation was given one by one for the pharmacokinetic of each drug administrated with and without CsA.

#### 2.3.1. DDI of CsA with Atorvastatin

Atorvastatin is a substrate of OATPs, CYP3A4, BCRP, P-gp and MRP2. Renal efflux transporters (BCRP, P-gp and MRP2) also mediated renal excretion of atorvastatin, although its contribution to total clearance was minor. Coadministration of CsA induced DDI with atorvastatin via inhibiting CYP3A4 and above transporters. Two clinical studies for DDIs of CsA and atorvastatin were included in the study.

Report 1 [14,16]. Eighteen renal transplant patients received CsA-based immunosuppressive therapy. The CsA dosage was 5.20 (2.03–12.66) mg/kg/d. The patients orally received daily 10 mg atorvastatin for four weeks. Pharmacokinetics of atorvastatin was performed following a four-week treatment. Eight age-matched healthy subjects were selected for controls and administrated daily 10 mg atorvastatin for one week [14]. The results showed that the simulated pharmacokinetic profile of atorvastatin following oral multi-dose of 40 mg atorvastatin was comparable to observed data (Figure 2A). Predicted and observed steady-state AUC_0–24_ and C_max_ of atorvastatin in healthy subjects and in renal transplant patients are show in Table 6. Dosages of CsA were set to be 886 mg/70 kg/d. The predictions were within the fold error range compared with clinical observations. The predicted AUCR and C_max_R values were in line with observations.

Report 2 [15]. Thirteen healthy subjects orally received 40 mg atorvastatin for six days. On the evening of day 6, the first dose (2.5 mg/kg) of CsA was administrated. On day 7, CsA (2.5 mg/kg) was co-administrated with atorvastatin in the morning. Pharmacokinetics of atorvastatin after morning dose was implemented on day 6 and day 7. CsA dosage was set to be 175 mg/70 kg. The pharmacokinetic profiles and corresponding pharmacokinetic parameters of atorvastatin before and after the second dose of CsA were predicted using the developed PBPK model (Appendix A). The results showed that the simulated pharmacokinetic profile of atorvastatin following oral multi-dose of 40 mg atorvastatin was comparable to observed data. Predicted steady-state AUC_0–6_ and C_max_ following multi-dose of atorvastatin to healthy subjects were within 0.5–2.0 folds of clinic observations (data are shown in Table 6). Coadministration of CsA increased plasma exposure to atorvastatin. The predicted AUC_0–6_ and C_max_ of atorvastatin following co-administration of CsA were lower than the clinic observations. The predicted AUCR and C_max_R were also lower than observed data.

#### 2.3.2. DDIs of CsA with Cerivastatin

Cerivastatin is a substrate of P-gp, BCRP, OATP1B1 and CYP2C8. CsA-mediated DDI with cerivastatin is attributed to inhibiting OATP1B1-mediated hepatic uptake and P-gp/BCRP-mediated intestinal efflux. A clinic report [17] showed that 12 renal transplant recipients received CsA-based immunosuppressive therapy. The dosage regimen of CsA ranged from 75 mg to 225 mg twice daily. The patients were administrated 0.2 mg cerivastatin daily for seven days. Pharmacokinetics of cerivastatin were implemented on day 1 and day 7. Twelve healthy subjects were administrated a single dose of cerivastatin, serving as the control group. The predicted C_max_ and AUC of cerivastatin following oral 0.2 mg of cerivastatin were good agreement with the measured data in healthy subjects. The dosage of CsA was assumed 225 mg/d. The predicted C_max_ and AUC of cerivastatin following oral 0.2 mg of cerivastatin to renal transplant recipients on day 1 and day 7 were fell within 0.5–2.0 folds of observations, shown in Figure 2B and Table 6. Compared with healthy subjects, the predicted AUCR on day 7 was 5.2, falling within 0.5–2.0 folds of observation (4.8), but predicted C_max_R was 2.3, less than 0.5-fold observations (5.0).

#### 2.3.3. DDIs of CsA with Pravastatin

Pravastatin is a substrate of MRP2 and OATPs. The drug mainly eliminates via MRP2-mediated biliary and MRP2-mediated renal secretion, indicating that CsA leads to DDI with pravastatin via affecting OATP-mediated hepatic uptake, MRP2-mediated intestinal efflux, MRP2-mediated biliary excretion and MRP2-mediated renal secretion. Two studies for DDIs of CsA with pravastatin were included in the study.

Report 1 [18]. Twenty-three stable kidney allograft recipients undergoing CsA-based immunosuppressive therapy received a single 20 mg oral daily dose of pravastatin for 28 days. Pharmacokinetics of pravastatin were implemented on day 1 and day 28 following pravastatin administration. CsA was administered as an oral capsule formulation twice daily, whose dosage was set to be 420 mg/d. The pharmacokinetic profiles of pravastatin following multidose of 20 mg to patients on day 1 and day 28 were predicted and compared with observations (Figure 2C). The C_max_ and AUC_0–24_ of pravastatin co-administrated with CsA were predicted in Table 6. The predictions fell within 0.5–2.0 folds of clinic observations. The mean values in Table 5 were normalized by 20 mg, as controls, AUCR and C_max_R on day 28 were calculated to 3.1 and 3.6, respectively, within the fold error range compared with predictions (3.7 and 4.1 respectively).

Report 2 [19]. Eleven heart-transplant recipients underwent CsA-based immunosuppressive therapy. The patients received a daily dose of 40 mg/d pravastatin for the first eight days, then was reduced to 10 mg/d, administered until day 29. Pharmacokinetics of pravastatin were carried out on day 1, 8 and 29. Pharmacokinetics of pravastatin in eight healthy subjects following a single dose of 60 mg pravastatin was also investigated. The dosage of CsA in patients was set to 400 mg/d. The predicted C_max_ and AUC_0–24_ values normalized by 10 mg pravastatin in healthy subjects were in line with clinic observations. The predicted 10 mg normalized C_max_ values of pravastatin co-administrated CsA on days 1, 8 and 29 were also within the fold error range compared with observations. The predicted 10 mg-normalized AUC_0–24_ values in heart-transplant on day 1, 8 and 29 were lower than observations. The predicted C_max_R fell within 0.5–2.0 folds observations but estimated AUCR were lower than observations.

#### 2.3.4. DDI of CsA with Rosuvastatin

Rosuvastatin is a substrate of P-gp, BCRP, MRP2 and OATPs. Rosuvastatin mainly eliminates via BCRP-dependent biliary and renal secretion. Thus, DDI of CsA with rosuvastatin may be due to inhibition of these transporter function. A report [20] demonstrated heart transplanted patients underwent a standard scheme of CsA-based therapy and 10 healthy subjects received a single dose of 10 mg rosuvastatin, followed by once-daily oral dose of 10 mg rosuvastatin for 10 days. Another group might receive a dose of 20 mg rosuvastatin, followed by a once-daily oral dose of 20 mg rosuvastatin for 10 days. Pharmacokinetics of rosuvastatin were documented at the first and last dose for patients and healthy subjects. CsA dosage was set to be 200 mg twice a day. The pharmacokinetics of rosuvastatin in healthy subjects and patients were predicted and compared with observation (Figure 2D and Table 6). The predicted C_max_ and AUC of rosuvastatin following multi-dose of 10 mg rosuvastatin to healthy subjects were agreed with observations. The predicted C_max_ and AUC of single-dose and multi-dose of 10 mg to patients were lower than that for measured geometric mean, the same for single-dose and multi-dose of 20 mg rosuvastatin to patients. The predicted AUCR and C_max_R following multi-doses to patients were 2.4 and 2.9, respectively, which were also lower than observations.

#### 2.3.5. DDIs of CSA with Fluvastatin

Fluvastatin is metabolized by CYP2C9. Fluvastatin is also a substrate of P-gp, BCRP, MRP2 and OATPs. Liver BCRP also mediates biliary excretion of fluvastatin. Thus DDI of CsA with fluvastatin mainly results from inhibiting the above transporters. Two clinic reports were cited.

Report 1 [21]. Ten heart transplant patients under CsA-based therapy and 10 healthy subjects received 40 mg fluvastatin daily for 28 days. Pharmacokinetics was documented on day 1 and day 28. CsA dosage was assumed to be 200 mg/d. The plasma concentrations of fluvastatin in patients and healthy subjects were predicted and compared with observations (Figure 2E and Table 6). The predicted C_max_ and AUC_0–24_ on day 1 and day 28 following oral doses of 40 mg fluvastatin to healthy subjects were within the 2-fold error range compared with the observations. The predicted C_max_ and AUC_0–24_ on day 1 and day 28 following oral 40 mg of fluvastatin to heart transplant patients were also fell within 0.5-2.0 folds of clinical observations. On day 28, predicted C_max_R and AUCR were 3.4 and 3.2, respectively, which were in line with observations (6.0 and 3.1, respectively).

Report 2 [92]. Twenty hypercholesterolemic renal transplant recipients receiving CsA in combination with azathioprine and methylprednisolone administrated 20 mg fluvastatin daily for 4–6 weeks. CsA dosage was set to 200 mg/d. The predicted C_max_ value was falling within 0.5–2.0 folds of observation, but the predicted AUC was twice higher than observation.

#### 2.3.6. DDI of CsA with Simvastatin

Simvastatin is a substrate of OATPs and CYP3A4, which mainly eliminates via the CYP3A4-mediated metabolism mechanism. CsA leads to DDI with simvastatin via inhibiting OATP-mediated hepatic uptake, CYP3A4-mediated metabolism in the intestine and liver. Two clinic reports were included in the simulation.

Report 1 [93]. Pharmacokinetics of simvastatin following an oral dose of 20 mg was implemented in five transplant patients receiving CsA (1.1~3.8 mg/kg) and 5 patients without CsA. The dosage of CsA was set to be 266 mg/70 kg. The pharmacokinetic profiles of simvastatin following oral 20 mg simvastatin to patients treated with CsA or without CsA were predicted and compared with observations (Figure 2F and Table 6). The predicted AUC_0–24_ and C_max_ values in patients treated with CsA were fell within ranges of observed AUC_0–24_ and C_max_. The predicted AUC_0–24_ and C_max_ values in patients treated without CsA were lower than clinic observations. Predicted AUCR and C_max_R were 3.6 and 7.3, respectively, which were higher than observations (2.7 and 2.08). However, using observed mean values normalized by 20 mg in Table 5, AUCR values of C_max_ and AUC were calculated to 6.5 and 6.9, respectively, within the 2-fold error range compared with predictions (7.3 and 3.6. respectively).

Report 2 [94]. Seven hypercholesterolemic heart transplant patients under CsA-based immunosuppressive therapy and seven hypercholesterolemic non-heart transplant patients (served as the control) were included. Both groups undergoing 10 mg simvastatin treatment for 6 weeks, attended the clinic in the morning after an overnight fast and received a double dose (20 mg). The dosage of CsA was set to be 200 mg twice daily. The predicted concentrations of simvastatin acid at 1, 2 and 3 h following oral 20 mg of simvastatin to heart transplant patients were higher than those in non-heart transplants. The observed mean plasma concentrations of simvastatin acid in heart transplant patients at 1, 2, and 3 h following oral 20 mg simvastatin were higher than those in non-heart transplants. The plasma concentrations of simvastatin acid in non-heart transplants were lower than 2 ng/mL. The predictions were higher than the observations.

#### 2.3.7. DDI of CsA with Lovastatin

Lovastatin is a substrate of OATPs, MRP2 and CYP3A4, which mainly eliminates via CYP3A4-mediated metabolism mechanism. DDI of CsA with lovastatin occurs via inhibiting OATP-mediated hepatic uptake, MRP2-mediated intestinal efflux and CYP3A4-mediated metabolism. Three clinic reports were cited in the study.

Report 1 [18]. Twenty-one stable kidney allograft recipients undergoing CsA-based immunosuppressive therapy received a single 20 mg oral daily dose of lovastatin for 28 days. CsA dosage was between 2 and 6 mg/kg/day. Pharmacokinetic profiles of lovastatin were implemented on day 1 and day 28. CsA was administered as an oral capsule formulation twice daily. Pharmacokinetic profiles of lovastatin co-administrated 420 mg CsA were simulated and compared with observations (Figure 2G and Table 6). The predicted C_max_ and AUC of lovastatin on day 1 and day 28 were lower than the observed medium values. Using the data list in Table 5 as controls, the calculated C_max_R and AUCR were within the 2-fold error range compared with predicted.

Report 2 [95]. Five types of patients were investigated as follows: (1) six heart transplant recipients receiving CsA (273 mg)-based triple immunosuppressive regimen; (2) five kidney transplant recipients receiving CsA (230 mg)-based triple immunosuppressive regimen; (3) five patients with psoriasis treated with CsA (290 mg) monotherapy; (4) five kidney transplant recipients receiving prednisolone and azathioprine; and (5) eight hypercholesterolemic patients serving as “control”. These patients orally received 10 mg lovastatin once daily for 10 days. Pharmacokinetics of lovastatin was implemented on day 10 (Appendix A). The simulated AUC_0–8_ values in heart transplant recipients receiving CsA, kidney transplant recipients receiving CsA, patients with psoriasis treated with CsA, and hypercholesterolemic patients were 31.33, 27.87, 32.64, and 3.89 ng·h/mL. The measured mean AUC_0–8_ values in heart transplant recipients receiving CsA, kidney transplant recipients receiving CsA, patients with psoriasis treated with CsA and hypercholesterolemic patients were 175, 110, 110 and 26 ng·h/mL. Compared with hypercholesterolemic patients, AUCRs of lovastatin following co-administration of CsA were calculated to be 6.7, 4.2 and 4.2, respectively. The predicated AUCRs of lovastatin in heart and kidney transplant recipients and patients with psoriasis were 8.1, 7.2 and 8.4, respectively. All predictions were fell within 0.5–2.0 folds of observations.

Report 3 [96]. Twenty-four (14 cardiac and 10 renal) transplanted patients under CsA based-triple immunosuppressive regimen, received lovastatin treatment. Dosage of CsA was 300 (150–400) mg/d for heart transplanted patients and 255 (200–400) mg/d for renal transplanted patients. Five milligrams lovastatin was then given once daily in the morning for three weeks. The dose was thereafter increased by 5 mg every third week until 20 mg once daily, which was continued until week 18. Blood levels of lovastatin were measured prior to and 2 h after oral CsA and the first dose of lovastatin at each dose level. Blood levels of lovastatin in hypercholesterolaemic non-transplanted patients prior to and 2 h after 10 mg lovastatin were served as control. The dosages of CsA were set to be 300 mg. The predicted concentrations of lovastatin at 2 h following lovastatin 5 mg at weeks 0 and 3, 10 mg at week 6, 15 mg at week 9, and 20 mg at weeks 12 and 18 were 5.09, 5.93, 12.27, 18.91, 24.77 and 25.70 ng/mL, respectively. They were within 0.5–2.0-fold of observations. The predicted concentrations at 2 h following lovastatin 10 mg to hypercholesterolaemic non-transplanted patients was 1 ng/mL, lower than the observation concentration 4.6 ng/mL. Predicted AUCR using concentration at 2 h following 10 mg to transplanted patients at 6 weeks was 12, which was higher than observation (4.6).

#### 2.3.8. DDI of CsA with Repaglinide

Repaglinide eliminates mainly via CYP2C8. To some extent, CYP3A4 also mediates repaglinide metabolism. Repaglinide is also a substrate of OATP1B1 and P-gp, indicating that DDI of CsA with repaglinide is involved in the inhibition of CYP3A4, P-gp, and OATPs.

A report [23] showed that 12 healthy subjects at PM 8 on day 1 and at 8 AM on day 2, received 100 mg CsA or placebo. At 9 a.m. on day 2, they ingested a 0.25 mg dose of repaglinide. Pharmacokinetics of repaglinide was documented. The plasma concentrations of repaglinide following 0.25 mg dose of repaglinide alone or coadministration of CsA were simulated and compared with observations (Figure 2H and Table 6). The predicted C_max_ and AUC of repaglinide when co-administrated CsA were 2.4- and 2.5-fold higher than those following repaglinide alone. The measured C_max_ and AUC of repaglinide when co-administrated CsA was 1.7 folds and 2.4 folds higher than those following repaglinide alone. All predictions fell within 0.5–2.0 folds of observations.

#### 2.3.9. DDI of CsA with Bosentan

Bosentan is mainly eliminated via the metabolic mechanism, which is partly attributed to CYP3A4. Bosentan is also a substrate of MRP2 and OATPs and renal MRP2 may be involved in renal excretion of bosentan. These results inferred that DDI of CsA with bosentan is attributed to inhibition of CYP3A4-mediated metabolism, OATPs-mediated hepatic uptake and MRP2-mediated renal secretion.

A report [24] demonstrated that bosentan was given to 10 healthy subjects twice daily at a dose of 500 mg for 8 days. The subjects received 300 mg CsA twice daily starting from day 1 evening for 8 days. On days 1 and 8, pharmacokinetics of bosentan was documented following the morning dose. The plasma concentrations of bosentan were stimulated (Figure 2I and Table 6). The predicted C_max_ and AUC of bosentan following coadministration of CsA were 2.2-fold and 3.76-fold higher than those before co-administration of CsA. The measured C_max_ and AUC of bosentan following coadministration of CsA were higher than those before coadministration of CsA. Predicted C_max_R and AUCR were within the fold error range compared with the observed values.

### 2.4. Sensitivity Analysis of Model Parameters

Hepatic clearance of these tested agents is highly dependent on OATP-mediated uptake [26]. The empirical scaling factor (SF) is often introduced to gaps in translating in vitro transporter kinetic data. The K_i.OATP_ of CsA and clinic dosage of CsA largely varied. Five parameters, SF, CL_int,uptake_, K_i_,_OATP_, CsA dosage and transition rate (K_t_) on DDIs of CsA with atorvastatin were investigated. Variabilities of K_i_,_OATP_ SF, CL_int,uptake_, CsA dosage and K_t_ were set to be 0.5, 1 and 2.

We set atorvastatin as a typical drug to investigate the contribution of five parameters mentioned above. With the same structure of the PBPK model, we have reason to believe that other substances may with a similar effect by these factors. CsA dosage was set to be 200 mg/d. The results showed that the tested four factors remarkably affected DDIs of CsA with atorvastatin (Figure 3A–F), whose extents were CL_int,uptake_ ≈ SF > CsA dosage > K_i,OATP_ > K_t_. Individual contributions of hepatic CYP3A, integrated effects of the liver (i.e., hepatic CYP3A4-mediated metabolism/hepatic OATP-mediated hepatic uptake) and integrated effects of the intestine (i.e., intestinal CYP3A-mediated metabolism/efflux transporters) to DDI of CsA with atorvastatin were also investigated (Figure 3F). The results demonstrated that no considering inhibitions of CsA on CYP3A-mediated metabolism (non-CYP3A4) and integrated effects occurring in the liver (non-liver) or intestine (non-gut) led to remarkably lower concentrations of atorvastatin compared with actual concentrations of atorvastatin when co-administrated CsA. These results indicated that DDI of CsA with atorvastatin should be attributed to integrated effects of CYP3A4, OATP, P-gp, MRP2 and BCRP inhibitions. The pharmacokinetics parameters variation made by different factors listed in Figure 3 are shown in Table 7 (DDI Part).

Atorvastatin was used as mode drug to further investigate individual contributions of CYP3A, OATPs, P-gp, MRP and BCRP to atorvastatin disposition (Figure 4). The results showed that individual contributions to atorvastatin disposition following an oral dose were hepatic OATPs >> hepatic CYP3A > intestinal CYP3A ≈ intestinal P-gp ≈ intestinal MRP2 ≈ intestinal BCRP. The integrated contribution of intestinal CYP3A, intestinal P-gp, intestinal MRP2 and intestinal BCRP to atorvastatin disposition following oral dose contributions was about 20%. The pharmacokinetics parameters variation made by different factors listed in Figure 4 are shown in Table 7 (Alone Part).

## 3. Discussion

Drug disposition is highly dependent on the interplay of drug metabolism enzymes (such as CYP450 and UGTs) and transporters (such as OATPs, P-gp, BCRP and MRP2). These enzymes and transporters are widely expressed in the liver, intestine, or kidney. Some drugs, such as atorvastatin, are substrates of these enzymes and transporters, some of which (such as CsA) are also inhibitors of these enzymes and transporters. When they were co-administered, DDIs occur due to the inhibition of these enzymes and transporters. The extents of DDIs should be attributed to integrated effects on enzymes and transporters in these tissues. The aim of the study was to establish a PBPK model characterizing the interplay of enzymes and transporters in the intestine, liver and kidney, to predict DDIs of CsA with victim drugs.

Nine victim drugs were selected. They are all substrates of OATPs, some of which are also substrates of CYP3A4, BCRP, P-gp or MRP2. The developed PBPK model was first validated using nine victim drugs and perpetrator CsA using in vitro parameters of drug metabolism and transport. The lag time of simvastatin can be interpreted as the time for simvastatin (lactone) converse to simvastatin acid. The results showed that the developed PBPK model was successfully applied to predict pharmacokinetic profiles of nine victim drugs and CsA. Of predications, 96% pharmacokinetics parameters fell within 0.5–2.0 folds of observation. Following validation, the developed PBPK model was successfully scaled to DDIs of CsA with nine victim drugs. Except for rosuvastatin, all DDI predictions had the same trend with observations, demonstrating that the CsA-mediated DDIs may be predicted using the developed PBPK model involving enzymes and transporters in the liver, intestine and kidney.

Atorvastatin was served as a model drug for sensitivity analysis and individual contributions of enzymes and transporters to drug disposition. A sensitivity analysis that CL_int,up_, K_i,OATP_, SF, the dosage of CsA and intestinal transit remarkably affected DDI of CsA with atorvastatin. The effect of SF, CL_int,up_ was the strongest, followed by CsA dosage and K_i,OATP_. The K_i_ values for CsA were reported to vary 6.3-folds when different OATP1B1 substrates were used in the in vitro transporter inhibition assays [97], showing a possible substrate-dependent inhibition of cyclosporine on OATP1B1 functions. In the clinic, the CsA dosage was varied, which may explain large variances in plasma concentrations of victim drugs when co-administered CsA. The empirical scaling factor (SF) was frequently applied during PBPK model development to capture the observed in vivo clearance [98]. The present study showed that each of the substrates had itself SF values and 300-fold varied (from CsA 0.1 to atorvastatin 32.6) among the tested agents. For a special substrate, the reported SF value was often different. For example, during simulating pharmacokinetics of atorvastatin. We also estimated SF using data from 10 reports. The results showed SF values had 5.4 times variations (ranging from 10.6 to 57). The medium value (32.6) was selected for simulating atorvastatin disposition, and good predictions were obtained. All these indicated that unreasonable SF value may lead to mispredictions. The ratio of drug concentration in tissue to plasma (K_p_), which is used to describe drug distribution, is one of the most important factors for the PBPK model. We used a different method to get K_p_ value and selected the most reasonable one to continue our research showed in Table 2.

The simulation also demonstrated that intestinal BCRP, P-gp, MRP2 and CYP3A, hepatic OATPs and CYP3A differently contributed to atorvastatin disposition, hepatic OATPs > hepatic CYP3A > intestinal CYP3A ≈ intestinal BCRP ≈ intestinal P-gp ≈ intestinal MRP2. Systemic exposure of atorvastatin was more affected by hepatic OATP1B1 inhibition than by CYP3A4 inhibition, supporting that OATP1B1 was the rate-limiting step for the hepatic clearance of atorvastatin. All these results also demonstrated that DDI of CsA with atorvastatin should be mainly attributed to hepatic OATP inhibition, which may explain the findings that the exposure of atorvastatin increased 12-fold in the presence of the OATP1B1 inhibitor rifampin, but did not change in the presence of the strong CYP3A4 inhibitor itraconazole [99]. Stimulation also demonstrated that contribution of intestinal integrated effects (CYP3A-mediated metabolism and P-gp/BCRP/MRP2-mediated efflux) to atorvastatin disposition following oral dose was about 20%, leading to low intestinal bioavailability. High exposure to CsA following an oral dose of CsA occurred, inferring that extent of DDI following oral administration was larger than that following intravenous dose

However, the study also has some limitations. DDIs of CsA with statins were mainly documented in organ transplant patients under CsA-based therapy. The predictions of DDIs using the PBPK model were dependent on the assumption that diseases little affected pharmacokinetics of the indicated drugs. In fact, these diseases and the co-medicine themselves may affect pharmacokinetics. For example, rosuvastatin is mainly eliminated via the kidney. Heart failure was reported to also contribute to the decline of kidney function [100], which seemed to explain under- prediction of plasma exposure to rosuvastatin in heart transplanted patients. Moreover, OATP1B1 polymorphisms also affect the pharmacokinetics of statins [101] and repaglinide [101]. All these may lead to mispredictions, which needed further investigation.

In conclusion, the PBPK model characterizing the interplay of enzymes and transporters was successfully applied to predict the pharmacokinetics of OATP substrates and DDIs of CsA with 9 victim drugs. Most of the predictions fell within 0.5–2.0-fold of observations.

## 4. Method

### 4.1. Collection of Data

Clinical DDI studies involving CsA and OATP substrates were collected from publications on Pubmed. The CsA-mediated DDIs were based on the following criterions: (1) data might come from healthy subjects or patients following a single-dose or multi-dose of victim drugs when co-administrated with CsA. What is more, DDI data and the data for drugs used alone might come from different reports. (2) CsA was orally administrated to subjects in an immediate-release formulation. (3) Pharmacokinetic parameters such as peak concentration (C_max_) or area under the curve (AUC) or pharmacokinetic profiles were shown.

### 4.2. Development of a Whole-Body PBPK Model Charactering Interplay of Enzymes and Transporters in Intestine, Liver and Kidney

A whole-body PBPK model characterizing the interplay of enzymes and transporters in the intestine, liver and kidney (Figure 5) was developed to predict pharmacokinetic behaviors of victim drugs and CsA as well as DDIs of CsA with victim drugs. The PBPK model consisted of the lung, heart, brain, muscle, kidney skin, spleen, adipose tissues, other body tissues, gastrointestinal systems and liver [26,50,102]. 

In gastrointestinal systems

Gastrointestinal systems were divided into the stomach, gut lumen and gut wall. Assuming that absorption or metabolism of drug did not occur in stomach, drug amount (A_0_) in stomach was controlled by transition rate (K_ti_), i.e.,
(1)$$\frac{{\mathrm{dA}}_{0}}{\mathrm{dt}}=\;-{\;K}_{\mathrm{ti}}\times A_{0}$$
where the ith stand for different parts of gastrointestinal systems (i = stomach/duodenum/jejunum/ileum/cecum/colon).

The gut lumen was divided into five segments including duodenum, jejunum, ileum, caecum and colon according to their physiological and anatomical characteristics. Drug absorption was assumed to only occur at duodenum, jejunum and ileum.

The disposition of the drug in the ith gut lumen (i = duodenum/jejunum/ileum) was illustrated by
(2)$$\frac{{\mathrm{dA}}_{i}}{\mathrm{dt}}=K_{\mathrm{ti}\;-\;1}\times A_{i\;-\;1}\;-{\;K}_{\mathrm{ti}}\times A_{i}\;-{\;k}_{a,i}\times A_{i}+k_{b,i}\times A_{\mathrm{gwi}}\times T_{\mathrm{sf},i}$$
where K_ti_, k_a,i_ and A_gwi_ were gut transit rate constant, absorption rate constant in the ith gut segment and the drug amount in the ith enterocyte compartment. k_b,i_ is the efflux rate constant of the drug from enterocytes into the lumen. The expressions of intestinal P-gp, BCRP and MRP2 are regional [103], thus, transporter-mediated P_eff, B-A_ in the ith gut segment was corrected by a relative transporter scaling factor (T_sf,i_). The expressions of P-gp, BCRP and MRP2 in ileum were assumed to be 1. According to a previous report [103], the T_sf,i_ values of P-gp, BCRP and MRP2 in duodenum, jejunum and ileum were calculated as follows: 0.16:1.2:1.0 for P-gp, 0.68:1.45:1.0 for BCRP and 1.27:1.77:1.0 for MRP2. k_a,i_ and k_b,i_ may be calculated from human effective permeability in apical to basolateral direction (P_eff,man,A-B_) and in basolateral to apical direction (P_eff,man,B-A_) across the intestinal wall, i.e.,
(3)$$k_{a,i}=2\times P_{\mathrm{eff},A-B}/r_{i}$$
(4)$$k_{b,i}=2\times P_{\mathrm{eff},\;B-A}/r_{i}$$
where r_i_ is the intestinal radius and the r_i_ values for duodenum, jejunum and ileum were set to be 2.0, 1.63 and 1.45 cm [102], respectively. P_eff, B-A_ was determined by intestinal P-gp, BCRP and MRP2, which should be attributed to the summed effect of these efflux transporters (P-gp, BCRP and MRP2). In the presence of CsA, P_eff, B-A_ was rewritten as
(5)$$P_{\mathrm{eff},B-A}=\frac{P_{\mathrm{eff},B-A,P-\mathrm{gp}}}{1+(A_{\mathrm{gwi}}^{I}/V_{\mathrm{gwi}})/K_{i,P-\mathrm{gp}}}+\frac{P_{\mathrm{eff},B-A,\mathrm{BCRP}}}{1+(A_{\mathrm{gwi}}^{I}/V_{\mathrm{gwi}})/K_{i,\mathrm{BCRP}}}+\frac{P_{\mathrm{eff},B-A,\mathrm{MRP}2}}{1+(A_{\mathrm{gwi}}^{I}/V_{\mathrm{gwi}})/K_{i,\mathrm{MRP}2}}$$
where V_gwi_ and $A_{\mathrm{gwi}}^{I}$ are the volume of the ith intestinal segment and amount of inhibitor in the ith intestinal segment. P_eff, B-A, P-gp,_ P_eff, B-A, BCRP,_ and P_eff, B-A, MRP2_ are efflux parameters mediated by P-gp, BCRP and MRP2, respectively. K_i, P-gp_, K_i, BCRP_ and K_i, MRP2_ are inhibition constants of corresponding transporters by CsA.

The drug amount (A_gwi_) in the ith gut wall is expressed as:(6)$$\frac{{\mathrm{dA}}_{\mathrm{gwi}}}{\mathrm{dt}}{=\;Q}_{\mathrm{gwi}}\times \frac{A_{\mathrm{art}}}{V_{\mathrm{art}}}{\;+\;k}_{a,i}{\times A}_{i}{\;-\;k}_{b,i}{\times A}_{\mathrm{gwi}}{\times T}_{\mathrm{sf},i}{\;-\;(Q}_{\mathrm{gwi}}+\frac{f_{\mathrm{ub}}{\;\times \;\mathrm{CL}}_{\mathrm{int},\mathrm{gwi}}}{{1\;+\;(A}_{\mathrm{gwi}}^{I}{/V}_{\mathrm{gwi}}{)/K}_{i}}{)\times A}_{\mathrm{gwi}}{/V}_{\mathrm{gwi}}{/K}_{g:b}$$
where Q_gwi_ is the blood flow of the ith gut wall. A_art_, V_art_ and K_g:b_ are the amounts of drug in artery blood, volume of artery blood and ratio of drug concentrations in intestine to blood, respectively. CL_int, gwi_ is intrinsic clearance in the ith gut wall. Superscript “I” indicated perpetrator, CsA. The expression of intestinal CYP3A is also regional, expressions of CYP3A4 in duodenum, jejunum and ileum were 9.7 nmol, 38.4 nmol and 22.4 nmol [53], respectively. CL_int,gut_ values of atorvastatin, CsA, simvastatin and lovastatin in intestinal microsomes were reported to be 0.03, 0.0277, 1.86 and 2.143 mL/mg microsome protein, respectively. From the report [104], there were 3155.2 mg microsomal protein/intestine for humans. According to the volume of the intestinal segment [105], median regional wet weights of duodenum, jejunum and ileum were 79 (0.098), 411 (0.39) and 319 (0.51) g, respectively. Assume that intestinal metabolic activity mainly comes from the jejunum. CL_int, jejunum_ was calculated as 0.51 × 3155.2 × CL_int, gut_. While, with the reported expression of regional CYP3A4 [53], convert jejunal metabolism to CYP3A metabolism and we can get CL_int, duodenum_ and CL_int, ileum_, which are listed in Table 4.

In artery blood (art):(7)$$\frac{{\mathrm{dA}}_{\mathrm{art}}}{\mathrm{dt}}{=\;Q}_{\mathrm{total}}{\times ((A}_{\mathrm{lu}}{/V}_{\mathrm{lu}}{)/K}_{\mathrm{lu}:b}{-A}_{\mathrm{art}}{/V}_{\mathrm{art}})$$
where Q_total_, V_lu_, A_lu_ and K_lu:b_ were cardiac output, lung volume, amount of drug in lung and ratio of drug concentration in lung to blood, respectively.

In venous blood (ven):(8)$$\frac{{\mathrm{dA}}_{\mathrm{ven}}}{\mathrm{dt}}=\sum Q_{i}{\times (A}_{i}{/V}_{i}{)/K}_{\mathrm{tissue}:b}{-\;Q}_{\mathrm{total}}{\;\times \;A}_{\mathrm{ven}}{/V}_{\mathrm{ven}}$$
where K_tissue:b_ is ratio of drug concentration in different tissues to blood.

In liver:

The liver compartment was divided into hepatic blood and hepatocytes. Hepatic uptake of drugs was mainly controlled by OATPs and biliary excretion of drug was mediated by P-gp, BCRP or MRP2.

In hepatic blood (A_h, b_):(9)$$\frac{{\mathrm{dA}}_{h,b}}{\mathrm{dt}}=\sum \frac{Q_{\mathrm{gwi}}{\times A}_{\mathrm{gwi}}}{V_{\mathrm{gwi}}{\times K}_{g:b}}+\frac{Q_{\mathrm{sp}}{\times A}_{\mathrm{sp}}}{V_{\mathrm{sp}}{\times K}_{s:b}}+\frac{Q_{\mathrm{ha}}{\times A}_{h,b}}{V_{h}}\;-\;\frac{Q_{h}{\times A}_{h,b}}{V_{h}}-\;\left(\frac{{\mathrm{SF}}_{\mathrm{act}}{\times \mathrm{CL}}_{\mathrm{int},\mathrm{up},\mathrm{OATP}}}{1+\frac{f_{\mathrm{ub}}^{I}{\times A}_{h,b}^{I}}{V_{h,b}{\times K}_{i,\mathrm{OATP}}}}{+\mathrm{CL}}_{\mathrm{int},\mathrm{pd}}\right)\;{\times \;f}_{\mathrm{ub}}\times \frac{A_{h,b}}{V_{h,b}}{+f}_{\mathrm{ub}}{\times \mathrm{CL}}_{\mathrm{int},\mathrm{pd}}\times \frac{A_{h,c}}{V_{h,c}{\times K}_{h:b}}$$
where V_h, b_, V_h, c_, K_h:p_ and Q_h_ (Q_h_ = Q_ha_ + Q_sp_ + Q_gwi_) are the volume of hepatic blood and hepatocytes, ratio of drug concentrations in liver to blood, and blood flow in liver venous. CL_int, up, OATP_ and CL_int, pd_ are OATP-mediated intrinsic clearance of uptake and passive clearance, respectively. SP indicates the spleen. SF_act_ represents the empirical scaling factor which was applied to scale-up the in vitro clearance to the in vivo hepatic clearance to make up for the underpredicted overall hepatic intrinsic clearance [27]. SF_act_, the value of atorvastatin, cerivastatin, pravastatin, rosuvastatin, repaglinide and bosentan, was cited from reference, while SF_act_, the value of cyclosporine, fluvastatin, simvastatin and lovastatin, was estimated from the software based on their pharmacokinetic data, which were shown in Appendix A. At least three sets of clinical reports were used for SF_act_ estimation, and their means were used for PBPK simulation. The estimated and cited SF_act_ values are listed in Table 3.

In hepatocytes (A_h, c_):(10)$$\begin{array}{ll} \frac{{\mathrm{dA}}_{h,c}}{\mathrm{dt}}= & \left(\frac{{\mathrm{SF}}_{\mathrm{act}}{\times \mathrm{CL}}_{\mathrm{int},\mathrm{up},\mathrm{OATP}}}{{1+f}_{\mathrm{ub}}^{I}{\;\times A}_{h,b}^{I}/V_{h,b}{/K}_{i,\mathrm{OATP}}}{+\mathrm{CL}}_{\mathrm{int},\mathrm{pd}}\right)\times \;\frac{f_{\mathrm{ub}}{\times A}_{h}}{V_{h}}\\ & -({\mathrm{CL}}_{\mathrm{int},\mathrm{pd}}+\frac{{\mathrm{CL}}_{\mathrm{int},\mathrm{bile}}}{{1+f}_{\mathrm{ub}}^{I}{\times A}_{h,c}^{I}/V_{h,c}{/K}_{i,P-\mathrm{gp}/\mathrm{BCRP}/\mathrm{MRP}2}}+\frac{{\mathrm{CL}}_{\mathrm{int},\mathrm{met},\mathrm{CYP}3A}}{{1+f}_{\mathrm{ub}}^{I}{\times A}_{h,c}^{I}/V_{h,c}{/K}_{i,\mathrm{CYP}3A}}\\ & {+\mathrm{CL}}_{\mathrm{int},\mathrm{met},\mathrm{non}-\mathrm{CYP}3A})\;\times \;\frac{f_{\mathrm{ub}}{\times A}_{h,c}}{V_{h,c}{\times K}_{h:b}} \end{array}$$
where CL_int, met, CYP3A_ and CL_int, met, non-CYP3A_ are intrinsic clearances of CYP3A-mediated metabolism and non-CYP3A-mediated metabolism, respectively. CL_int, bile_ is the biliary excretion clearance by P-gp, BCRP or MRP2. K_i, OATP_, K_i, CYP3A_ and K_i, P-gp/BCRP/MRP2_ are inhibition constants of OATP-mediated uptake, CYP3A-mediated metabolism and P-gp, BCRP-or MRP2-mediated biliary excretion by CsA, respectively.

In kidney:

Renal excretion of drug mainly occurs via glomerular filtration and renal secretion, the renal clearance (CL_ren_) is illustrated by
(11)$${\mathrm{CL}}_{\mathrm{ren}}{\;=\;f}_{\mathrm{ub}}\times {\mathrm{GFR}\;+\;\mathrm{CL}}_{\sec}$$
where GFR (set to be 120 mL/min/70 kg) and CL_sec_ are glomerular filtration rate and renal secretory clearance by P-gp, BCRP or MRP2, respectively. CL_sec_ was calculated using the equation:(12)$${\mathrm{CL}}_{\sec}{\;=\;\mathrm{CL}}_{\mathrm{ren}}{\;-\;f}_{\mathrm{ub}}\times \mathrm{GFR}$$

Assuming that DDI occurred at secretion process, amount of drug in kidney is
(13)$$\frac{{\mathrm{dA}}_{k}}{\mathrm{dt}}{\;=\;Q}_{k}\times \frac{A_{\mathrm{art}}}{V_{\mathrm{art}}}-\;\frac{A_{k\;}{\times \;f}_{\mathrm{ub}}}{V_{k}{\times K}_{k:b}}{\times (\mathrm{CL}}_{\mathrm{int},\mathrm{GFR}}+\frac{{\mathrm{CL}}_{\mathrm{int},\sec}}{1+\frac{f_{\mathrm{ub}}^{I}\times A_{k}^{I}}{V_{k}{\times K}_{i,P-\mathrm{gp},\mathrm{BCRP},\mathrm{MRP}2}}})$$

The CL_int,GFR_ and CL_int, sec_ were calculated using equation [106]
(14)$${\mathrm{CL}}_{\mathrm{GFR}}\left({\mathrm{or}\;\mathrm{CL}}_{\sec}\right)=\;\frac{f_{\mathrm{ub}}{\times \mathrm{CL}}_{\mathrm{int},\;\mathrm{GFR}}{(\mathrm{or}\;\mathrm{CL}}_{\mathrm{int},\;\sec}{)\;\times \;Q}_{k}}{f_{\mathrm{ub}}{\times \mathrm{CL}}_{\mathrm{int},\;\mathrm{GFR}}\left({\mathrm{or}\;\mathrm{CL}}_{\mathrm{int},\;\sec}\right){\;+\;Q}_{k}}$$
where Q_k_ is blood flow in kidney and CL_GFR_ = f_ub_ × GFR. The CL_ren_ of cyclosporine A and bosentan were 3.44 and 2.4 mL/min. CL_int, GFR_ of pravastatin and rosuvastatin were 132.9 mL/min. CL_int,sec_ of pravastatin (MRP2) and rosuvastatin (BCRP) were 1538.96 mL/min and 420.3 mL/min, respectively.

The amount (A_i_) of drug in other tissue is illustrated by the following formulation.
(15)$$\frac{{\mathrm{dA}}_{i}}{\mathrm{dt}}{=Q}_{i}\times (\frac{A_{\mathrm{art}}}{V_{\mathrm{art}}}-\frac{A_{i}}{V_{i}{\times K}_{\mathrm{tissue}:b}})$$
where Q_i_ is blood flow in other tissues.

Phoenix WinNonlin 8.1 (Pharsight, St. Louis, MO, USA) was used for coding and solving of the PBPK model as well as estimation of kinetic parameters (C_max_ and AUC).

### 4.3. Model Validation

The criteria for selecting victim drugs: 1. the victim drugs selected should be used clinically at present; 2. they all must be substances of OATP, some of which are substances of CYP3A; 3. they all must have clinical data of interactions with CsA, some of which may have strong DDI effect with CsA. Only the ones that meet all conditions were selected. The selected drugs may be combined with cyclosporine in certain patients clinically.

There were two steps in this process. Pharmacokinetic profiles and their parameters (AUC and C_max_) of nine victim drugs and CsA following oral administration to humans were first predicted using the developed PBPK models and compared with clinic observations. Following validation, the developed PBPK model was separately used to predict DDIs of CsA with victim drugs and confirmed with clinic data. The dosage of CsA was set to be the highest dosage reported in clinic reports. The extent of DDI was assessed as the ratio of AUC (AUCR) or C_max_ (C_max_R) with CsA to without CsA and absolute average fold error (AAFE) around each data point. The predictions were considered successful if the ratio of predication to observation and AAFE fell between 0.5 and 2.0.
(16)$$\mathrm{AAFE}=\;10^{\frac{1}{n}\sum_{i=1}^{n}\left|\log_{10}\frac{C_{\mathrm{Pred}}}{C_{\mathrm{Obs}}}\right|}.$$

### 4.4. Sensitivity Analysis of Model Parameters

Hepatic uptakes of the tested victims are mainly mediated by OATPs. Several pieces of evidence showed that the inhibitory effects of cyclosporin A were long-lasting [107], whose K_i_ showed about a 100-fold difference between pre-incubation non-pre-incubation. Moreover, the clinical dosage of CsA often varied. During in vitro–in vivo extrapolation of transporter-mediated clearance, the empirical scaling factor (SF) is frequently introduced to capture the observed in vivo clearance [98]. Sensitivity analysis was operated on CL_int,up_, K_i,OATP_, SF, CsA dosage and rate of intestinal transit.

Atorvastatin is a substrate of OATPs, P-gp, BCRP and CYP3A. CsA is a strong inhibitor of OATPs, P-gp, BCRP, MRP2 and CYP3A. The individual contributions of BCRP, P-gp, MRP2, CYP3A, OATPs to DDI of atorvastatin with CsA were investigated.

## Figures and Tables

**Figure 1 ijms-21-07023-f001:**
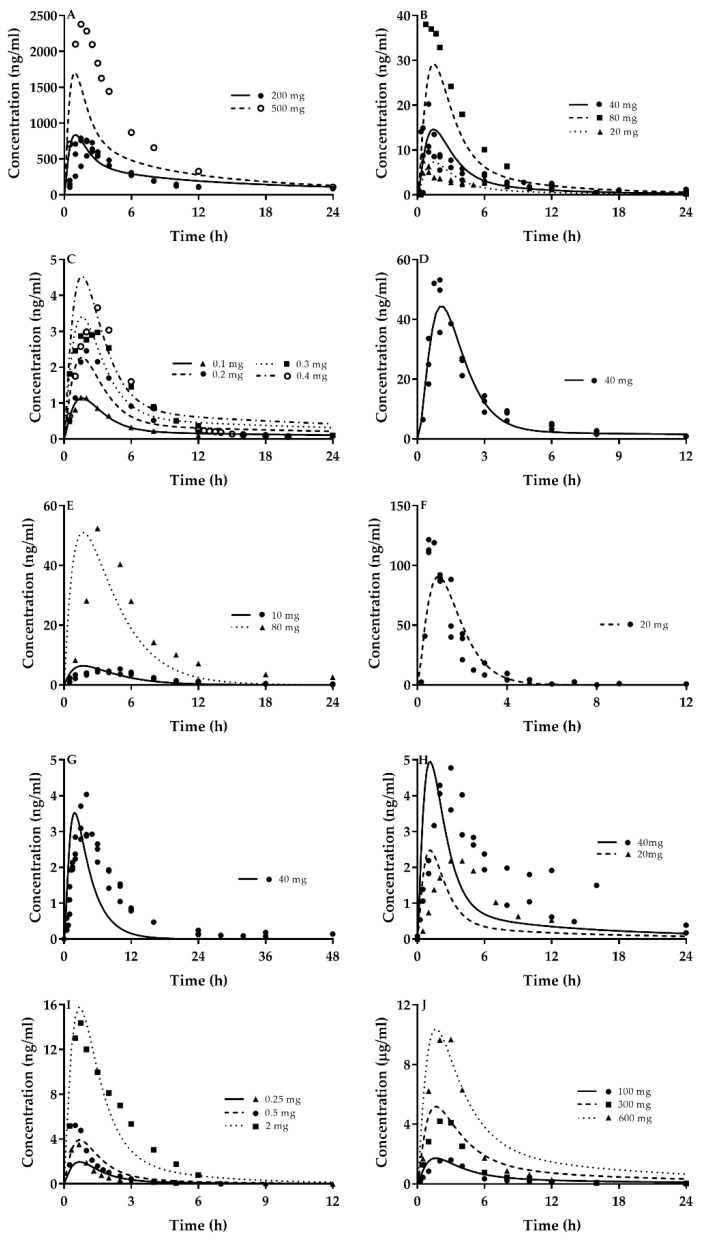
Predicted plasma concentrations (line) of the tested agents using the PBPK model and observed plasma concentrations (points) following a single dose to healthy subjects. Different points represented different dosages for drugs in the corresponding figure. (**A**) CsA; (**B**) atorvastatin; (**C**) cerivastatin; (**D**) pravastatin; (**E**) rosuvastatin; (**F**) fluvastatin; (**G**) simvastatin; (**H**) lovastatin; (**I**) repaglinide; (**J**) bosentan. Observed data were cited from reports [10,20,23,64,65,66,67,68,69,70,71,72,73,74,75,76,77,78,79,80,81,82,83,84,85,86,87,88,89,90].

**Figure 2 ijms-21-07023-f002:**
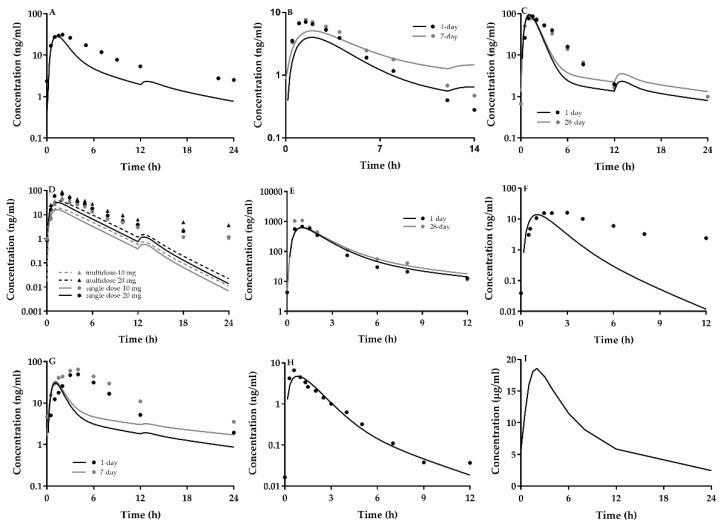
Predicted plasma concentrations (line) of the victim drugs agents using the PBPK model and observed plasma concentrations (points) following oral administration to subjects when co-administered with CsA (twice daily). (**A**) atorvastatin; (**B**) cerivastatin; (**C**) pravastatin; (**D**) rosuvastatin; (**E**) fluvastatin; (**F**) simvastatin; (**G**) lovastatin; (**H**) repaglinide; (**I**) bosentan. Observed data were cited from clinic reports [14,15,16,17,18,19,20,21,23,24,92,93,94,95,96].

**Figure 3 ijms-21-07023-f003:**
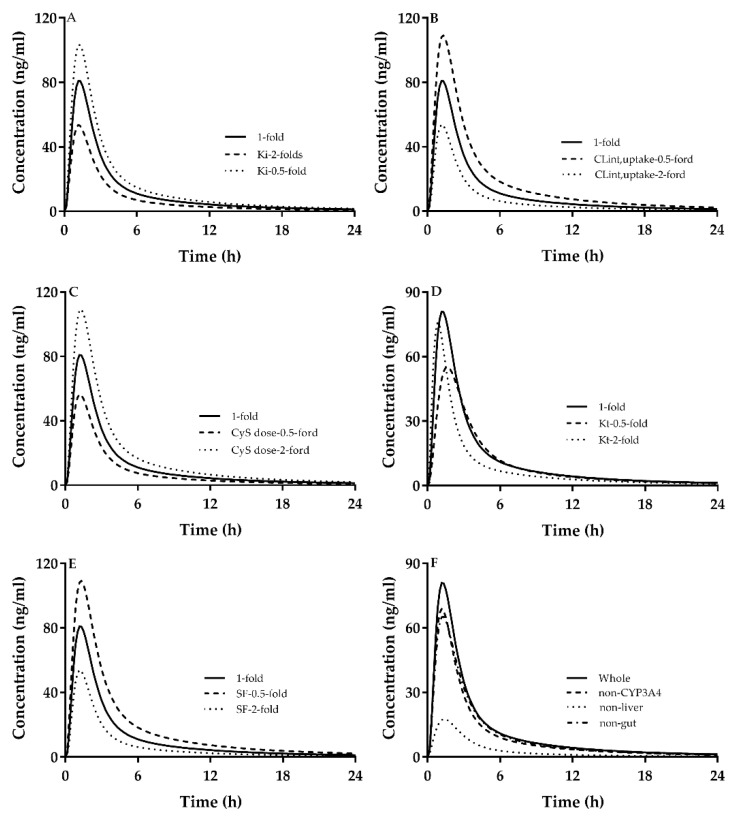
Contributions of (**A**) inhibition constant on hepatic organic anion transporting polypeptides (OATP) by CsA (K_i,OATP_), (**B**) OATP-mediated hepatic uptake (CL_int,up_), (**C**) CsA dosage (100 mg, 200 mg and 400 mg), (**D**) transition rate (K_t_) and (**E**) empirical scaling factor (SF) to plasma concentrations of atorvastatin following oral 40 mg of atorvastatin. CsA dosage was set to be 200 mg/d. Contributions of (**F**) inhibitions on CYP3A4 (non-CYP3A4), integrated effects of hepatic CYP3A-mediated metabolism/hepatic OATP-mediated uptake (non-liver) and integrated effects of intestinal CYP3A4-mediated metabolism/transporter-mediated intestinal efflux (non-gut) to the increased plasma concentrations of atorvastatin by CsA.

**Figure 4 ijms-21-07023-f004:**
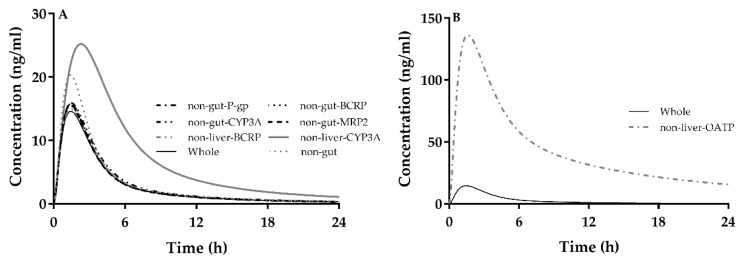
Effects of (**A**) hepatic CYP3A4 (non-liver CYP3A4), hepatic BCRP (non-liver BCRP), intestinal CYP3A4 (non-gut CYP3A4), intestinal P-gp (non-gut P-gp), intestinal BCRP (non-gut-BCRP), intestinal MRP2 (non-gut-MRP2) and integrated effects of the intestine (intestinal CYP3A4, intestinal P-gp, intestinal MRP2 and intestinal BCRP, non-gut) to plasma concentrations of atorvastatin following oral 40 mg of atorvastatin, which was compared with actual concentrations (Whole), (**B**) hepatic OATPs (non-liver-OATP).

**Figure 5 ijms-21-07023-f005:**
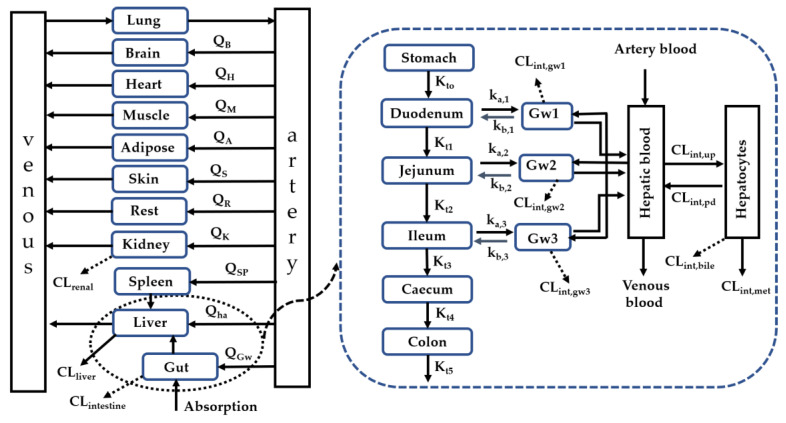
Schematic diagram of the whole PBPK model involving both enzyme and transporter turnover. Q_i_ indicated blood flow in the corresponding compartment. CL is the intrinsic clearance. G_wi_ is the the i-th enterocyte compartment. K_ti_, k_a,i_ and k_b,i_ represented the transit rate constant, drug absorption rate constant and efflux rate constant from enterocyte to gut lumen, respectively. CL_int, up_, CL_int, pd_, CL_int, bile_, CL_int, gwi_ and CL_int,met_ were uptake clearance, passive clearance, efflux clearance to bile, metabolism clearance in the gut and metabolic clearance in hepatocytes.

**Table 1 ijms-21-07023-t001:** Physiological parameters for humans used in the physiologically based pharmacokinetic PBPK model.

Tissue	V (mL) [38,39]	Q (mL/min) [38]	Transit Rate Constant (min^−1^) [40]
Lung	1170	5600	/
Kidney	280	1240	/
Heart	310	240	/
Liver	1690	1518	/
Muscle	35,000	750	/
Skin	7800	300	/
Brain	1450	700	/
Adipose	10,000	260	/
Rob	5100	592	/
Spleen	190	80	/
Artery	1730	5600	/
Venous	3470	5600	/
Stomach	160	38	0.0462
Duodenum	70	118	0.0462
Jejunum	209	413	0.012
Ileum	139	244	0.0058
Cecum	116	44	0.004
Colon	1116	281	0.0013

**Table 2 ijms-21-07023-t002:** Ratio of drug concentration in tissue to plasma for cyclosporin A (CsA), atorvastatin (Ato), cerivastatin (Cer), pravastatin (Pra), rosuvastatin (Ros), fluvastatin (Flu), simvastatin (Sim), bosentan (Bos), repaglinide (Rep) and lovastatin (Lov).

Tissue	CsA ^a^	Ato ^a^	Cer [41]	Pra ^a^	Ros [41]	Flu [42]	Sim [41]	Lov [43]	Rep [41]	Bos [41]
Adipose	253.35	181	0.43 ^a^	0.14	0.82	0.339	0.39	298.06	0.2	0.52
Liver	4.13 [44]	4.57	51	56 [45]	10.60	34.13	22.7	25.63	15.8	51
Muscle	3.67	2.64	0.2	0.21	0.20	0.147	0.219	4.69	0.302	0.23
Lung	7.75	5.6	1.13	0.36	0.40	0.407	0.426	9.06	0.604	1.01
Kidney	5.79	4.18	2.93	48.5 [45]	1.12	1.59	2.87	4.25	1.48	1.14
Brain	11.77	8.44	0.2	0.22	0.01	0.094	0.207	0.88	0.078	0.35
Heart	5.22	3.78	2	0.32	0.40	0.82	0.612	11.63	0.702	0.64
Intestine	5.99 [44]	8.97	0.2 ^a^	0.21	0.18 ^a^	8.2	20.2 ^a^	36.2	0.07 ^a^	0.47 ^a^
Skin	18.28	13.16	0.533	0.38	0.38	0.313	0.39	21.46	0.209	0.48
Spleen	3.36	2.43	0.733	0.26	0.23	0.23	0.292	0.31	0.346	0.52
Stomach	3.67	2.64	0.09	0.21	0.20 ^a^	1.22	20.2 ^a^	62.69	0.07 ^a^	0.16 ^a^
Rest	5.24	1	0.01	0.11	0.01	0.01	0.01	0.01	0.01	0.01
f_ub_ ^b^	0.05 [27]	0.084 [46]	0.017 [47]	0.56 [28]	0.174 [27]	0.014 [27]	0.107 [48]	0.03 [49]	0.011 [50]	0.02 [51]
R_b_ ^b^	1.36 [27]	0.61 [27]	0.76 [47]	0.839 [28]	0.69 [27]	0.57 [27]	0.56 [52]	0.57 [53]	0.62 [50]	0.6 [51]

^a^ calculated from PKa and logP [29]. ^b^ f_ub_ is unbound fraction of drug in blood. R_b_ is blood/plasma ratio.

**Table 3 ijms-21-07023-t003:** Intrinsic metabolic/transport parameters and empirical scaling factor (SF) of cyclosporin A (CsA), atorvastatin (Ato), cerivastatin (Cer), pravastatin (Pra), rosuvastatin (Ros), fluvastatin (Flu), simvastatin (Sim), lovastatin (Lov), repaglinide (Rep) and bosentan (Bos) in the liver.

Drug	CL_int met_		CL_int Uptake_	SF	CL_int, pd_	CL_int, bile_		
	CYP3A4	Other				P-gp	BCRP	MRP2
	mL/min		mL/min		mL/min	mL/min		
CsA	5432 [46]	/	10,857 [46]	0.1 ^a^	2933 [46]	637 [46]	/	/
Ato	3469.5 [27]	612.3 [27]	6374.7 [54]	32.6 [27]	3916.79 [46]	/	302.4 [55]	/
Cer	979.8 [27]	1197.6 [27]	1942.8 [56]	12.5 [27]	3541.5 [27]	40.5 [27]	/	/
Pra	/	/	283.3 [27]	19.4 [27]	80.9 [27]	/	/	80.9 [27]
Ros	/	/	1841.6 [27]	9.2 [27]	242.8 [27]	/	566.6 [27]	/
Flu	2386.3 [56]	2801.3 [56]	9106.6 [56]	21 ^a^	4047.4 [56]	/	3440 [56]	/
Sim	231,391 [57]	/	9234.1 [31]	25 ^a^	23,695.5 [31]	101.2 [31]	/	/
Lov	6893.96 [43]	/	13,129.8 [31]	4 ^a^	11,569.5 [31]	/	/	/
Rep	2503.7 [27]	6438 [27]	7184 [27]	16.9 [58]	4452.14 [27]	/	20.24 [27]	/
Bos	1365.14 [27]	/	7184 [27]	1.1 [27]	2023.7 [27]	1472.6 [51]	/	/

^a^ SF is the active uptake scaling factor from the simulation whose source has been shown in Appendix A.

**Table 4 ijms-21-07023-t004:** Intrinsic metabolic/transport parameters of cyclosporin A (CsA), atorvastatin (Ato), cerivastatin (Cer), pravastatin (Pra), rosuvastatin (Ros), fluvastatin (Flu), simvastatin (Sim), lovastatin (Lov), repaglinide (Rep) and bosentan (Bos) in the intestine.

Drug	k_a_	P_eff, A-B_	P_eff, B-A_			CL_int, gut_ [53]		
	min ^−1^	cm/min		cm/min		mL/min		
			P-gp	BCRP	MRP2	Duodenum	Jejunum	Ileum
CsA	0.025 [28]	/	0.01 [28]	/	/	11.25	44.6	25.98
Ato	/	0.0094 [59]	0.0185 [8]	0.0118 [8]	0.011 [8]	12.2	48.273	28.22
Cer	/	0.0133 [47]	0.0099 [60]	0.0065 [60]	0.0118 [60]	/	/	/
Pra	0.021 [28]	/	/	/	43.7 ^a^	/	/	/
Ros	0.0022 [61]	/	0.012 [8]	0.013 [8]	0.011 [8]	/	/	/
Flu	/	0.038 [62]	0.035 [8]	0.026 [8]	0.029 [8]	/	/	/
Sim	/	0.01 [49]	0.0093 [49]	/	/	755.2	2991.3	1745
Lov	/	0.016 [49]		/	0.014 [49]	871	89.8	2011
Rep	0.04 [58]		0.0148 [63]			/	/	/
Bos	/	0.014 [51]			5.84 ^b^ [51]	/	/	/

^a^ The unit of value is mL/min for the efflux of MRP2. ^b^ The value with the unit mL/min was calculated from V_max_ and K_m_.

**Table 5 ijms-21-07023-t005:** The predicted (Pred) pharmacokinetic parameters and the mean clinic observations (Obs) of cyclosporin A (CsA), atorvastatin (Ato), cerivastatin (Cer), pravastatin (Pra), rosuvastatin (Ros), fluvastatin (Flu), simvastatin (Sim), lovastatin (Lov), repaglinide (Rep) and bosentan (Bos).

Drug	Ref.	Dose	C_max_		Pred/Obs	AUC		Pred/Obs	AAFE
		mg	ng/mL			ng·h/mL			
			Pred	Obs		Pred	Obs		
CsA	[64]	200	681.17	947.16	0.72	5456.12	7113.47	0.77	1.5
	[64]	200	681.17	758.86	0.90	5456.12	6771.87	0.81	1.5
	[64]	200	681.17	938.71	0.73	5456.12	7253.95	0.75	1.6
	[65]	500	1702.93	2480	0.69	11,984.7	16,100	0.74	1.7
Ato	[66]	40	14.59	19.5	0.75	71.69	82.8	0.87	2.3
	[67]	40	14.59	20.22	0.72	71.77	87.48	0.82	1.8
	[68]	40	13.5	9.45	1.43	73.18	89.66	0.82	2.1
	[69]	40	14.59	11.6	1.26	66.70	69.4	0.96	1.6
	[70]	20	6.75	4.7	1.44	35.78	47.4	0.75	2.5
	[71]	20	7.29	8.9	0.82	32.65	34.5	0.95	1.7
	[72]	20	7.29	9.28	0.79	34.22	43.7	0.78	1.7
	[73]	20	6.75	7.87	0.86	37.25	39.9	0.93	1.7
	[74]	80	29.18	38.07	0.77	136.39	182.96	0.75	1.5
	[75]	80	29.18	88.9	0.33	143.5	302.5	0.47	1.9
Cer	[76]	0.1	1.13	1.28	0.88	6.06	6.49	0.93	1.3
	[76]	0.2	2.26	2.61	0.87	13.13	13.1	1.00	1.2
	[77]	0.3	3.4	3.2	1.06	21.07	20.9	1.01	1.7
	[76]	0.4	4.53	3.66	1.24	26.25	24.6	1.07	2.3
Pra	[78]	40	50.07	53.23	0.95	129.86	133.63	0.97	1.3
	[79]	40	50.07	36.9	1.37	127.98	93.69	1.37	1.4
	[80]	40	50.07	49.5	1.02	118.62	109	1.09	1.2
Ros	[20]	10	6.3	4.58	1.38	34.04	40.1	0.85	2.1
	[81]	10	6.3	5.8	1.09	34.04	45.9	0.74	2.6
	[81]	80	50.42	53.5	0.94	272.75	397	0.69	2.0
Flu	[82]	20	91.03	121.6	0.75	264.62	199.5	1.33	2.7
	[82]	20	91.03	112.9	0.81	264.62	192.5	1.37	2.1
	[83]	20	91.03	119.06	0.76	247.18	144.62	1.71	1.9
Sim	[84]	40	3.87	11.5	0.34	16.48	26.5	0.62	4.2
	[85]	40	3.87	4.3	0.90	16.48	29.3	0.56	7.7
	[86]	40	3.87	3.37	1.15	16.48	32.50	0.51	4.1
Lov	[87]	40	4.91	5.00	0.98	20.20	29.2	0.69	2.7
	[82]	20	2.46	2.2	1.12	10.44	15.5	0.67	1.9
	[88]	40	4.91	4.06	1.21	22.40	34.1	0.66	2.6
Rep	[89]	2	15.7	15.28	1.03	31.19	34.77	0.90	1.4
	[23]	0.25	1.97	3.99	0.49	4.15	4.44	0.93	1.4
	[10]	0.5	3.92	5.22	0.75	7.96	6.36	1.25	1.5
Bos	[90]	100	1689.97	1786	0.95	11,365.74	8180	1.39	1.9
	[90]	300	5074.94	5000	1.01	33,714.30	18,450	1.83	2.8
	[90]	600	10,163.50	9987	1.02	67,476.63	41,480	1.63	2.6

**Table 6 ijms-21-07023-t006:** The predicted (Pred) pharmacokinetic parameters and the mean clinic observations (Obs) of atorvastatin (Ato), cerivastatin (Cer), pravastatin (Pra), rosuvastatin (Ros), fluvastatin (Flu), simvastatin (Sim), lovastatin (Lov), repaglinide (Rep) and bosentan (Bos) combined with cyclosporine (CsA).

Drug.	Dose	CsA Dose	C_max-with_		C_max-without_		C_max_R		AUC_-with_		AUC_-without_		AUCR	
	mg	mg	ng/mL		ng/mL				ng·h/mL		ng·h/mL			
			Pred	Obs	Pred	Obs	Pred	Obs	Pred	Obs	Pred	Obs	Pred	Obs
Ato	10 [14,16]	886 BID 28-day	28.5	37.3	3.31	3.5	8.61	10.66	123.22	226	17.21	26	7.16	8.69
	40 [15]	175	74.3	362	14.6	26.5	5.10	13.66	204.78	1026	49.84	67	4.11	15.31
Cer	0.2 [17]	225 BID 1 day	4.02	7.8	2.26	1.56	1.78	5.00	27.4	36.2	9.5	9.53	2.88	3.80
		225 BID 7-day	5.11	7.82	2.26	1.56	2.26	5.01	49.27	45.3	9.5	9.53	5.19	4.75
Pra	20 [18]	420 BID 1 day	91.2	84	25.35	23.27	3.60	3.61	211.33	249	62.74	66.9	3.37	4.44
		420 BID 28-day	95.44	80	25.35	23.27	3.76	3.44	243.02	241	62.74	66.9	3.87	4.30
	10 [19]	400 BID 1 day	53.96	96	12.68	13.7	4.26	7.00	126.39	307.05	31.37	26.25	4.03	11.70
		400 BID 8-day	55.82	98	12.68	13.7	4.40	7.15	141.86	303.52	31.37	26.25	4.52	11.56
		400 BID 29-day	56.3	115.1	12.68	13.7	4.44	8.40	143.95	345.9	31.37	26.25	4.59	13.18
Ros	10 [20]	200 BID 1 day	15.86	39.8	6.3	4.58	2.52	8.69	62.56	197	34	40.1	1.84	4.91
		200 BID 10-day	18.47	48.7	6.3	4.58	2.93	10.63	81.9	284	34	40.1	2.41	7.06
	20 [20]	200 BID 1 day	31.7	66.5	\	\	\	\	125.11	308	\	\	\	\
		200 BID 10-day	36.94	83.0	\	\	\	\	163.81	424	\	\	\	\
Flu	40 [21]	200 BID 1 day	622.96	869.4	182.06	211.9	3.42	4.10	1667.4	1948.8	494.4	549.4	3.37	3.55
		200 BID 28-day	646.87	1530	190.14	254.7	3.40	6.01	2062.23	2615.3	640	841.8	3.22	3.11
	20 [92]	200 BID 28-day	314	155	\	\	\	\	1056	373	\	\	\	\
Sim	20 [93]	266 BID	13.91	20.6	1.9	9.9	7.32	2.08	28.84	101	8.08	39.6	3.57	2.55
	10 [94]	1h 200 BID ^a^	18.77	6.8	1.65	2 ^b^	11.38	3.40						
		2 h 200 BID ^a^	11.56	8.4	1.83	2 ^b^	6.32	4.20						
		3 h 200 BID ^a^	4.56	12.1	1.44	2 ^b^	3.17	6.05						
Lov	20 [18]	420 BID 1 day	30.58	46	2.46	2.2	12.43	20.91	113.98	243	10.44	15.5	10.92	15.68
		420 BID 28-day	33.12	75	2.46	2.2	13.46	34.09	164.89	459	10.44	15.5	15.79	29.61
	10 [95]	273 BID 10-day ^c^							31.33	175	3.89	26	8.05	6.73
		230 BID 10-day ^c^							27.87	110	3.89	26	7.16	4.23
		290 BID 10-day ^c^							32.64	110	3.89	26	8.39	4.23
	5 [96]	300 ^d^	5.09	4.05	1	4.6	5.09	0.88						
	5	300 ^d^	5.93	11.47	1	4.6	5.93	2.49						
	10	300 ^d^	12.27	21.08	1	4.6	12.27	4.58						
	15	300 ^d^	18.91	25.25	1	4.6	18.91	5.49						
	20	300 ^d^	24.77	19.79	1	4.6	24.77	4.30						
	20	300 ^d^	25.7	19.05	1	4.6	25.70	4.14						
Rep	0.25 [23]	100	4.64	6.7	1.97	3.9	2.36	1.72	10.38	10.82	4.15	4.44	2.50	2.44
Bos ^e^	500 [24]	300 BID	18.55	7.92	8.47	4.74	2.19	1.67	189.49	48.9	50.44	24.78	3.76	1.97

^a^ The concentration of simvastatin acid showed in C_max_ rows are measured at point 1, 2 and 3 h. ^b^ The observed plasma concentrations of simvastatin acid without were lower than 2 ng/mL. ^c^ Different doses of cyclosporine were given to different patients. See detail in Section 2.3.7 “DDI of CsA with lovastatin”. ^d^ The concentrations in C_max_ rows were at point 2 h following lovastatin 5 mg at weeks 0 and 3, 10 mg at week 6, 15 mg at week 9, and 20 mg at weeks 12 and 18. Control group was given lovastatin 10 mg. ^e^ The unit of Bosentan for C_max_ and AUC were μg/mL and μg·h/mL, respectively.

**Table 7 ijms-21-07023-t007:** Pharmacokinetics parameters of different factors on plasma concentrations of atorvastatin following 40 mg orally.

Factors	T_max_	C_max_	AUC
	h	ng/mL	ng·h/mL
DDI Part			
Whole	1.17	75.99	264.82
K_i_ (2-fold)	1.17	53.65	185.75
K_i_ (0.5-fold)	1.17	103.21	378.30
CL_int,up_ (2-fold)	1.17	48.13	154.69
CL_int,up_ (0.5-fold)	1.33	107.23	419.24
CsA dosage (100 mg)	1.17	51.48	178.84
CsA dosage (400 mg)	1.33	106.77	392.31
K_t_ (2-fold)	0.83	82.94	228.64
K_t_ (0.5-fold)	1.67	60.31	266.78
SF (2-fold)	1.17	48.13	154.69
SF (0.5-fold)	1.33	107.23	419.24
non-CYP3A4	1.17	68.91	240.63
non-liver	1.33	19.23	77.52
non-gut	1.17	55.96	214.06
Alone Part			
Whole	1.50	14.59	66.88
non-liver CYP3A4	2.33	25.23	172.17
non-liver BCRP	1.50	14.96	69.11
non-gut CYP3A4	1.50	15.91	74.17
non-gut P-gp	1.50	15.76	69.35
non-gut-BCRP	1.50	15.48	68.67
non-gut-MRP2	1.33	15.63	68.80
non-gut	1.33	20.36	82.16
non-liver-OATP	1.67	136.13	1077.21

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
