# Peer review of "Prediction of Cyclosporin-Mediated Drug Interaction Using Physiologically Based Pharmacokinetic Model Characterizing Interplay of Drug Transporters and Enzymes"

_ijms, 2020, doi:10.3390/ijms21197023_

Round 1
Reviewer 1 Report
Overall, the design and purpose of the experiment do not seem clear. And above all, the core of this study is the development of the method, but the thorough verification of the developed method was not performed. It is thought that thorough research needs to be carried out again with a clearer purpose and motivation. In this paper, the authors describe "a whole body PBPK model characterizing interplay of enzymes and transporters in intestine, liver and kidney to predict disposition of CsA and 9 victim drugs including atorvastatin, cerivastatin, pravastatin, rosuvastatin, fluvastatin, simvastatin, lovastatin, repaglinide and bosentan as well as DDIs of CsA with 9 victim drugs". This manuscript is very unclear and incomplete. It is thought that considerable improvement will be required to improve the manuscript. Major comments for the PBPK model: 1) Crucially, the presented PBPK model (Figure 5) does not meet the total blood flow balance, especially in gut and liver compartment with respect to the values in Table 1. What is the QLa in Figure 5? At line 448, recheck and define the equation (Qh = Qpv + Qsp + Qgwi). 2) The presented PBPK model simply incorporated the interplay of enzymes and transporters in intestine only. However, among 9 victim drugs, biliary and/or renal excretion play very important roles in their disposition in the body. Thus, the PBPK model dose not fully reflect the disposition characteristics of 9 victim drugs and neither lead to the proper interpretation for DDI with CsA. 3) The calculated transit time from stomach to colon based on the values in Table 1 by 1/transit rate constant was about 31 hr. This 31 hr is too high for normal human. 4) As we know, the ratio of drug concentration in tissue to plasma is very important factor for the PBPK model. The relative magnitude of this ratio to the organ blood flow is the key factor for the development and construction of PBPK model. Discuss this relative magnitude in liver and kidney with high ratio (Kp) victim drugs based on the presented PBPK model. Minor comments. There are many typos. English editing service will be required.Author Response
Dear reviewers:
Thanks for your comments, which are all valuable and very helpful for revising and improving our paper, as well as the important guiding significance to our researches. We have studied comments carefully and have made correction which we hope meet with approval. Revised portion are marked in red in the paper. The main corrections in the paper and the responds to the reviewer’s comments are as flowing:
Point 1:
The question for the design and purpose of the experiment, as well as the method used in this study.
Response:
Thanks for your advice. We are very sorry for our unclear expression. We rewritten “Introduction” as follows:
In “Introduction”
“There have been researches about the DDI between statin and CsA with PBPK model [1,2], but few have integrated simultaneously of liver, intestinal and kidney on drug disposal. The highlight of our work is to investigate the integrity effect of enzymes and transporters in liver, intestinal and kidney on drug disposition using a whole PBPK model characterizing interplay of enzyme-transporter in liver, intestine and kidney. The developed PBPK first was used to predict disposition of CsA and 9 victim drugs (OATP substrates) including atorvastatin, cerivastatin, pravastatin, rosuvastatin, fluvastatin, simvastatin, lovastatin, repaglinide and bosentan as well as DDIs of CsA with 9 victim drugs. The predictions were compared with clinic reports. Atorvastatin, substrate of uptake transporter (OATP), efflux transporter (BCRP, P-gp, MRP) and enzyme CYP3A, was represented to further investigate individual contributions of transporters and CYP3A4 to disposition of drugs and to document sensitivity analysis.”
Point 2:
This manuscript is very unclear and incomplete. It is thought that considerable improvement will be required to improve the manuscript.
Response:
Thanks for your advice. We have written abstract and introduction according to your advice.
In “Abstract”
“We aimed to establish a whole-body physiologically-based pharmacokinetic (PBPK) model which predict disposition of CsA and 9 victim drugs including atorvastatin, cerivastatin, pravastatin, rosuvastatin, fluvastatin, simvastatin, lovastatin, repaglinide and bosentan as well as DDIs of CsA with 9 victim drugs to investigate the integrity effect of enzymes and transporters in liver, intestinal and kidney on drug disposition.”
In “Introduction”
“There have been researches about the DDI between statin and CsA with PBPK model [1,2], but few have integrated simultaneously of liver, intestinal and kidney on drug disposal. The highlight of our work is to investigate the integrity effect of enzymes and transporters in liver, intestinal and kidney on drug disposition using a whole PBPK model characterizing interplay of enzyme-transporter in liver, intestine and kidney. The developed PBPK first was used to predict disposition of CsA and 9 victim drugs (OATP substrates) including atorvastatin, cerivastatin, pravastatin, rosuvastatin, fluvastatin, simvastatin, lovastatin, repaglinide and bosentan as well as DDIs of CsA with 9 victim drugs. The predictions were compared with clinic reports. Atorvastatin, substrate of uptake transporter (OATP), efflux transporter (BCRP, P-gp, MRP) and enzyme CYP3A, was represented to further investigate individual contributions of transporters and CYP3A4 to disposition of drugs and to document sensitivity analysis. ”
Point 3:
Crucially, the presented PBPK model (Figure 5) does not meet the total blood flow balance, especially in gut and liver compartment with respect to the values in Table 1. What is the QLa in Figure 5? At line 448, recheck and define the equation (Qh = Qpv + Qsp + Qgwi).
Response:
Thanks for your advice. We are very sorry for our incorrect writing. The QLa represent the blood flow in liver artery. It is our mistake that the abbreviations are not unified, which caused the misunderstanding of blood flow. We have rechecked and redefine the abbreviations. We used Qha to represent the blood flow in liver artery with value 300 mL/min and Qh to represent the blood flow in liver venous with the Qha value plus the spleen and intestinal contribution, total 1518 mL/min. The equation Qh = Qha + Qsp + Qgwi described the blood flow in liver venous. We have corrected them in the revised manuscript.
Point 4:
The presented PBPK model simply incorporated the interplay f enzymes and transporters in intestine only. However, among 9 victim drugs, biliary and/or renal excretion play very important roles in their disposition in the body. Thus, the PBPK model dose not fully reflect the disposition characteristics of 9 victim drugs and neither lead to the proper interpretation for DDI with CsA.
Response:
Thanks for your advice. We are sorry not to clearly illustrate them in Table 5. Among tested 10 OATP substrates, only bile clearance of lovastatin was not considered. Bile clearances of these compounds were mediated by P-gp (cyclosporine A, cerivastatin, simvastatin and bosentan), BCRP (atorvastatin, rosuvastatin, fluvastatin, and repaglinide) and MRP2 (pravastatin), respectively. Renal secretions clearance of rosuvastatin and pravastatin were mediated by renal BCRP and MRP2, respectively. We have re-arranged Table 5 and results.
In Table 5
In Results
“Renal efflux transporters (BCRP, P-gp and MRP2) also mediated renal excretion of atorvastatin, although its contribution to total clearance was minor.”
“Pravastatin is a substrate of MRP2 and OATPs. The drug mainly eliminates via MRP2-mediated biliary and MRP2-mediated renal secretion, indicating that CsA leads to DDI with pravastatin via affecting OATP-mediated hepatic uptake, MRP2-mediated intestinal efflux, MRP2-mediated biliary excretion and MRP2-mediated renal secretion.”
“Rosuvastatin is a substrate of P-gp, BCRP, MRP2 and OATPs. Rosuvastatin mainly eliminates via BCRP-dependent biliary and renal secretion.”
“Fluvastatin is metabolized by CYP2C9. Fluvastatin is also substrate of P-gp, BCRP, MRP2 and OATPs. Liver BCRP also mediates biliary excretion of fluvastatin. Thus, DDI of CsA with fluvastatin mainly results from inhibiting above transporters.”
“Bosentan mainly eliminates via metabolism mechanism, which partly is attributed to CYP3A4. Bosentan is also a substrate of MRP2 and OATPs and renal MRP2 may be involved in its renal excretion of bosentan. These results inferred that DDI of CsA with bosentan is attributed to inhibition of CYP3A4-mediated metabolism, OATPs-mediated hepatic uptake and MRP2-mediated renal secretion.”
Point 5:
The calculated transit time from stomach to colon based on the values in Table 1 by 1/transit rate constant was about 31 hr. This 31 hr is too high for normal human.
Response:
Thanks for your advice. It is really true as Reviewer suggested that 31 hr is too high for normal human. So we searched other literatures on PubMed. But the other literatures also showed the similar transit time in different part of gut, as 0.25 h (stomach), 0.26 h (duodenum), 0.84 h (jejunum), 0.43 h (ileum), 4.19 h (cecum), 12.57 h (colon) [3]. The transit time we used was from the literature which also used to build a successful PBPK model [4]. To see the transit time values, cecum and colon occupy most of the time, while which parts without material exchange and have no significant effect on drug disposition. After the above consideration, we want to keep the initial value.
Point 6:
As we know, the ratio of drug concentration in tissue to plasma is very important factor for the PBPK model. The relative magnitude of this ratio to the organ blood flow is the key factor for the development and construction of PBPK model. Discuss this relative magnitude in liver and kidney with high ratio (Kp) victim drugs based on the presented PBPK model.
Response:
Thanks for your advice. we have added this part in Discussion section. In fact, we had tried different method to get Kp value. Following figure showed the different result from different souse of Kp. We found the report got human Kp value by experiment [5] and used different method [6-8] to calculate Kp. We finally choose method-1 to continue our research. Although the curve from experiment Kp value was similar with that from method-1, but we found the Kpliver value got from experiment was too high to added in establishing.
Reference
- Varma, M.V.; Bi, Y.A.; Kimoto, E.; Lin, J. Quantitative prediction of transporter- and enzyme-mediated clinical drug-drug interactions of organic anion-transporting polypeptide 1B1 substrates using a mechanistic net-effect model. J Pharmacol Exp Ther 2014, 351, 214-223, doi:10.1124/jpet.114.215970.
- Varma, M.V.; Lai, Y.; Feng, B.; Litchfield, J.; Goosen, T.C.; Bergman, A. Physiologically based modeling of pravastatin transporter-mediated hepatobiliary disposition and drug-drug interactions. Pharm Res 2012, 29, 2860-2873, doi:10.1007/s11095-012-0792-7.
- Heikkinen, A.T.; Baneyx, G.; Caruso, A.; Parrott, N. Application of PBPK modeling to predict human intestinal metabolism of CYP3A substrates - an evaluation and case study using GastroPlus. Eur J Pharm Sci 2012, 47, 375-386, doi:10.1016/j.ejps.2012.06.013.
- Perdaems, N.; Blasco, H.; Vinson, C.; Chenel, M.; Whalley, S.; Cazade, F.; Bouzom, F. Predictions of metabolic drug-drug interactions using physiologically based modelling: Two cytochrome P450 3A4 substrates coadministered with ketoconazole or verapamil. Clin Pharmacokinet 2010, 49, 239-258, doi:10.2165/11318130-000000000-00000.
- Mikkaichi, T.; Nakai, D.; Yoshigae, Y.; Imaoka, T.; Okudaira, N.; Izumi, T. Liver-selective distribution in rats supports the importance of active uptake into the liver via organic anion transporting polypeptides (OATPs) in humans. Drug Metab Pharmacokinet 2015, 30, 334-340, doi:10.1016/j.dmpk.2015.06.003.
- Rodgers, T.; Rowland, M. Physiologically based pharmacokinetic modelling 2: predicting the tissue distribution of acids, very weak bases, neutrals and zwitterions. J Pharm Sci 2006, 95, 1238-1257, doi:10.1002/jps.20502.
- Schmitt, W. General approach for the calculation of tissue to plasma partition coefficients. Toxicol In Vitro 2008, 22, 457-467, doi:10.1016/j.tiv.2007.09.010.
- Ruark, C.D.; Hack, C.E.; Robinson, P.J.; Mahle, D.A.; Gearhart, J.M. Predicting passive and active tissue:plasma partition coefficients: interindividual and interspecies variability. J Pharm Sci 2014, 103, 2189-2198, doi:10.1002/jps.24011.

Reviewer 2 Report
Overall
The authors have presented an interesting study. It was well executed and contains a lot of information. The use of PBPK modelling to investigate the interactions of drugs is an elegant way. I would recommend a language review of the text.
Intro
No comments
M&M
What software was used to build and execute the PBPK model?
What software was used for the PK analysis of the data, e.g. AUC and Cmax?
Page 12, line 382-383: “4) DDI data might come from 382 different reports.” What does this mean? Which DDI data does this regard? Does it mean that the PK data of the victim drug can come from one report and then the PK data of the victim drug coadministered with CsA comes from another report?
Page 13, line 386: “A whole PBPK model” should be “A PBPK model” or “A whole-body PBPK model”.
Figure 5: Please add Hepatic blood to the empty box that represents the hepatic blood compartment.
Figure 5, legend: Please add the abbreviation Gw to the text.
Page 13, line399-404: is the transition rate through the gut compartments the same as the gastric emptying rate? And what is that rate? Please update the text.
Page 13, line 408: “The drug process of drugs in the ith gut lumen”, What does this mean? Is it the absorption process?
Page 15, lines 458-466: How can the GFR be estimated (eq 14) if it was set to 120 ml/min/70kg?
Are the equations used for the PBPK model based on other papers, or similar to other papers. In that case, please reference to these papers.
Which parameters are estimated? To which data was the model fitted to estimate these parameters? What was the result of these estimations? Throughout the paper parameter estimations are mentioned a couple of times. All these questions should be addressed in the method section, and the results of the estimations should be presented clearly of course including the variability of the estimation.
I suggest that Table 1, 2 and 3 are added to the method section, it would make the reading of the method section easier. This is then assuming that none of the parameters are estimated. A table with estimated parameters might be included in the results section.
Section “4.3 Model validation”.
To my understanding, there were 2 steps in the simulations. Firstly, simulations of the plasma concentration-time curves of the 9 victim drugs and CsA was performed. Secondly, the simulations of the plasma concentration-time curves of the 9 victim drugs coadministered with CsA were performed. Is this correctly interpreted?
Questions: Were any parameters estimated during the first phase? If so, then these are not predictions, but curve fittings.
What were the criteria for successful predictions in the first and second phase, 0.5-2 fold different between pred/obs of AUC and Cmax for both phases? If so, could you clarify that?
Are AUC and Cmax the best indicators of a good prediction of a curve? I suggest that the prediction quality of at least the first phase should be calculated as either (absolute) average fold error OR by median prediction error and mean absolute percentage error. This would show better whether the prediction follows the observed data.
Section “4.4 Sensitivity analysis of model parameters”:
Add the information on how much all parameters were changed.
Why was there a difference in how much the parameters were changed (Ki,OATP was different)
How were the results interpreted? Again with a ratio between AUC or Cmax. Please update the text with how the data was analyzed.
It is also unclear to me if the effect was only investigated with atorvastatine, or also other substances.
Results
Section 2.2 was good and well written. I appreciated the Table and Figure. The Table might be updated with the ratio pred/obs, to give an easy insight into which ones were outside of the range.
Section 2.3
Please include a table containing the results. I find that an easy overview of the results of the victim drug coadministered together with CsA would really increase the readability of the results section. Table could be similar with Table 4, but also include the CmaxR and AUCR (Obs, predicted, and the ratio obs/pred).
Please refer to the figure in each of the DDI sections, to make it easier to find the correct result.
Please add all results in figure form (for example, I am missing one atorvastatine report in Figure 2)
Page 4, line 35: Fig 3A, should be Fig 2A
Could Figure 2 have the x-axis in hours instead of minutes? In the whole text the administrations are described in hours, and it is easier to match the text with the figure that way.
Section 2.4
Could a table with the results be added? That will clarify the absolute changes the variation in parameter values makes.
Figure 4, it is difficult to see the difference between Whole and non-gut
Discussion
Please refrain from using phrases like “most of”. Please use phrases that mention the actual percentage/ratio/increase.
Page 11, line 336. As there was no table with the absolute values, it is difficult for the reader to verify this statement. But could it be that the effect of Ki,OATP was strongest because the parameter value was varied 5x more than the other parameters?
Author Response
Dear Reviewer:
Thanks for your comments, which are all valuable and very helpful for revising and improving our paper, as well as the important guiding significance to our researches. We have studied comments carefully and have made correction which we hope meet with approval. Revised portion are marked in red in the paper. The main corrections in the paper and the responds to the reviewer’s comments are as flowing:
Point 1
What software was used to build and execute the PBPK model? What software was used for the PK analysis of the data, e.g. AUC and Cmax?
Response:
Thanks for your advice. We have added the software in the revised manuscript according to your comments
In Method
“Phoenix WinNonlin 8.1 (Pharsight, St. Louis, MO) was used for coding and solving of the PBPK model as well as estimation of kinetic parameters (Cmax and AUC).”
Point 2:
Point 2a:
Page 12, line 382-383: “4) DDI data might come from 382 different reports.” What does this mean? Which DDI data does this regard?
Response:
Thanks for your advice. We are very sorry to misunderstand it. DDI data came from different reports. Not “382”. The 382” may be the line number we guess.
Point 2b
Does it mean that the PK data of the victim drug can come from one report and then the PK data of the victim drug coadministered with CsA comes from another report?
Response
Thanks for your advice. We are very sorry to not clearly illustrate it.The answer for this question is “Yes”. The observed PK data of drugs for single and combined administration came from different report.We have rewritten method in the revised manuscript as follows.
“4) DDI data and the data for drugs used alone might come from different reports”.
Point 3
Page 13, line 386: “A whole PBPK model” should be “A PBPK model” or “A whole-body PBPK model”.
Response:
Thanks for your advice. We have changed the describe from “A whole PBPK model” to “A whole-body PBPK model” in the whole manuscript according to your advice.
Point 4
Figure 5: Please add Hepatic blood to the empty box that represents the hepatic blood compartment.
Response:
Thanks for your advice. We have added “Hepatic blood” in the figure 5.
Figure 5. Schematic diagram of whole-PBPK model involving both enzyme and transporter turnover. Qi indicated blood flow in corresponding compartment. CL was intrinsic clearance. Gwi were the ith enterocyte compartment. Kti, ka,i and kb,i represented the transit rate constant, drug absorption rate constant and efflux rate constant from enterocyte to gut lumen, respectively. CLint, up, CLint, pd, CLint, bile, CLint, gwi and CLint,met were uptake clearance, passive clearance, efflux clearance to bile, metabolism clearance in gut and metabolic clearance in hepatocytes.
Point 5:
Figure 5, legend: Please add the abbreviation Gw to the text.
Response:
Thanks for your advice. We have added Gw in the figure caption as showing above.
Point 6:
Page 13, line 399-404: is the transition rate through the gut compartments the same as the gastric emptying rate? And what is that rate? Please update the text.
Response:
Thanks for your advice. We are very sorry for our inconsistent writing. The transition rate and the gastric emptying rate both have the same meaning. We have unified the definition Kt as constants of transition rate throughout the manuscript.
Point 7:
Page 13, line 408: “The drug process of drugs in the ith gut lumen”, What does this mean? Is it the absorption process?
Response:
Thanks for your advice. We are sorry not to clearly illustrate it. We have rewritten it
“The disposition of drug in the ith gut lumen (i=duodenum, jejunum, ileum) was illustrated by”
Point 8:
Page 15, lines 458-466: How can the GFR be estimated (eq 14) if it was set to 120 ml/min/70kg?
Response:
Thanks for your advice. We are sorry to lead to misunderstanding. We have rewritten method in the revised manuscript.
Assuming that DDI occurred at secretion process, amount of drug in kidney is
|
|
(13) |
The CLint,GFR and CLint, sec might be calculated using equation [1]
|
|
(14) |
Where Qk is blood flow in kidney and CLGFR=fub×GFR. .
Point 9:
Are the equations used for the PBPK model based on other papers, or similar to other papers. In that case, please reference to these papers.
Response:
Thanks for your advice. We have added the reference in the manuscript [2-4].
Point 10:
- a) Which parameters are estimated? b) To which data was the model fitted to estimate these parameters? c) What was the result of these estimations? d) Throughout the paper parameter estimations are mentioned a couple of times. All these questions should be addressed in the method section, and the results of the estimations should be presented clearly of course including the variability of the estimation.
Response:
Thanks for your advice. We are very sorry not to clearly state them. Several reports[5,6] showed that the overall hepatic intrinsic clearance (CLint,h) underpredicted the in vivo hepatic clearance of drugs, introduction of an empirical scaling factor for active uptake clearance (SFact) improved scaling -up the in vitro clearance to the in vivo hepatic clearance. Thus, SF was also introduced to the simulation.
- a) The only estimated parameters estimated to fit observed data is Empirical scaling factors (SFact).
- b) SF values of atorvastatin, cerivastatin, rosuvastatin, repaglinide and bosentan were cited from reference. SF values of cyclosporine, pravastatin, fluvastatin, simvastatin and lovastatin were estimated from the software based on their pharmacokinetic data.
- c) The source of the SF data had been shown in Table 3. Except the data came from other reports, the SF data estimated by software were based on the observed clinical plasma concentration data. One group of clinical data can be fitted to one SF value, so the SF presented in the table is a mean value from at least three reports. The equation used to estimate SF is showing following:
|
(9) |
- d) we have added this statement in Method section
“SFact represents empirical scaling factor which was applied to scale-up the in vitro clearance to the in vivo hepatic clearance to make up for the underpredicted overall hepatic intrinsic clearance [5]. SFact value of atorvastatin, cerivastatin, rosuvastatin, repaglinide and bosentan were cited from reference, while SFact value of cyclosporine, pravastatin, fluvastatin, simvastatin and lovastatin were estimated from the software based on their pharmacokinetic data. At least three sets of clinic reports were used for SFact estimation, their means were used for PBPK simulation. The estimated and cited SFact values were listed in Table 3.
Point 11
I suggest that Table 1, 2 and 3 are added to the method section, it would make the reading of the method section easier. This is then assuming that none of the parameters are estimated. A table with estimated parameters might be included in the results section.
Response:
Thanks for your advice. Parameters in Table 1 were from reference. Parameters in Table 2 were from reference or calculated based on reference. Parameters in Table 3 were from reference or estimated. We tried to remove the Table 1-3, but the journal has the rule with “Figures should be placed in the main text near to the first time they are cited”. And in result section, the data used to estimate had to mentioned. So the position of the table has not moved.
Point 12
To my understanding, there were 2 steps in the simulations. Firstly, simulations of the plasma concentration-time curves of the 9 victim drugs and CsA was performed. Secondly, the simulations of the plasma concentration-time curves of the 9 victim drugs coadministered with CsA were performed. Is this correctly interpreted?
Response:
Thanks for your advice and affirmation. This is correctly interpreted. We did follow these steps. Following these two steps, we did the sensitive analysis using atorvastatin as typical drug.
Point 13
Were any parameters estimated during the first phase? If so, then these are not predictions, but curve fittings.
Response:
Thanks for your advice. We are sorry not to clearly illustrate it. In the study, only empirical scaling factors (SFact) values of some drugs including cyclosporine, pravastatin, fluvastatin, simvastatin and lovastatin based on their pharmacokinetic data. At least three sets of clinic reports were used for SFact estimation, their means were used for PBPK simulation. SFact of other drugs were cited from reports. Several reports[5,6] showed that the overall hepatic intrinsic clearance (CLint,h) underpredicted the in vivo hepatic clearance of drugs, introduction of an empirical scaling factor for active uptake clearance (SFact) improved scaling -up the in vitro clearance to the in vivo hepatic clearance.
We have rewritten the method
“SFact represents empirical scaling factor which was applied to scale-up the in vitro clearance to the in vivo hepatic clearance to make up for the underpredicted overall hepatic intrinsic clearance [5]. SFact, value of atorvastatin, cerivastatin, rosuvastatin, repaglinide and bosentan were cited from reference, while SFact, value of cyclosporine, pravastatin, fluvastatin, simvastatin and lovastatin were estimated from the software based on their pharmacokinetic data. At least three sets of clinic reports were used for SFact estimation, their means were used for PBPK simulation. The estimated and cited SFact values were listed in Table 3.”
Point 14
What were the criteria for successful predictions in the first and second phase, 0.5-2 folds different between pred/obs of AUC and Cmax for both phases? If so, could you clarify that?
Response:
Thanks for your advice. Fold error was made to be the criteria for prediction successful or not. Fold errors of both AUC and Cmax between observed and predicted data were less than 2, indicating validity of the PBPK model in simulations for both phases. We have added this criterion in the manuscript as following:
“The sign of successful prediction is that fold errors of both AUC and Cmax between predicted and observed data were less than 2.”
Point 15
Are AUC and Cmax the best indicators of a good prediction of a curve? I suggest that the prediction quality of at least the first phase should be calculated as either (absolute) average fold error OR by median prediction error and mean absolute percentage error. This would show better whether the prediction follows the observed data.
Response:
Thanks for your advice. we have added absolute average fold error (AAFR) to illustrate the quality of prediction with AUC and Cmax in the first phase in Table 4.
Point 16: Add the information on how much all parameters were changed.
Response:
Thanks for your advice. Five parameters, SF, CLint,uptake, Ki,OATP, CsA dosage and gastric emptying rate (Kt) on DDIs of CsA with atorvastatin were investigated. We have added them in corresponding section as following.
“Five parameters, SF, CLint,uptake, Ki,OATP, CsA dosage and transition rategastric emptying rate (Kt) on DDIs of CsA with atorvastatin were investigated.”
Point 17
Why was there a difference in how much the parameters were changed (Ki,OATP was different)
Response:
Thanks for your advice. Because there are huge differences in Ki reported in different literature. The Ki values for CsA were reported to vary 6.3-folds when different OATP1B1 substrates were used in the in vitro transporter inhibition assays [7], as mentioned in Discussion. So we made is different.
Point 18
How were the results interpreted? Again with a ratio between AUC or Cmax. Please update the text with how the data was analyzed.
Response:
Thanks for your advice. we have added the Table 6 to describe the pharmacokinetics parameters (AUC and Cmax) in this section.
Point 19
It is also unclear to me if the effect was only investigated with atorvastatin, or also other substances.
Response:
Thanks for your advice. Sorry for our unclear statement. In this test, we made atorvastatin as a representative to investigate the effect. With the same structure of PBPK model, we have reason to believe that other substances may with the similar effect by these factors. We have added this statement as following in corresponding section:
“We set atorvastatin as a typical drug to investigate the contribution of five parameters mentioned above. With the same structure of PBPK model, we have reason to believe that other substances may with the similar effect by these factors.”
Point 20
Section 2.2 was good and well written. I appreciated the Table and Figure. The Table might be updated with the ratio pred/obs, to give an easy insight into which ones were outside of the range.
Response
Thanks for your advice. We have added the ratio pred/obs of Cmax and AUC in Table 4.
Point 21
Please include a table containing the results. I find that an easy overview of the results of the victim drug coadministered together with CsA would really increase the readability of the results section. Table could be similar with Table 4, but also include the CmaxR and AUCR (Obs, predicted, and the ratio obs/pred).
Response
Thanks for your advice. We have added the Table 5 to make the result clearer.
Point 22
Please refer to the figure in each of the DDI sections, to make it easier to find the correct result.
Response:
Thanks for your advice. It’s our negligence and we have added as suggested.
Point 23:
Please add all results in figure form (for example, I am missing one atorvastatin report in Figure 2)
Response:
Thanks for your advice. We have added figure cite to make it clear. In order to make the article description more concise and readable, we choose to put one pre/obs figure for each drug.
Point 24
Page 4, line 35: Fig 3A, should be Fig 2A
Response:
Thanks for your advice. We have corrected.
Point 25
Could Figure 2 have the x-axis in hours instead of minutes? In the whole text the administrations are described in hours, and it is easier to match the text with the figure that way.
Response
Thanks for your advice. We have changed the x-axis from minutes to hours.
Point 26
Could a table with the results be added? That will clarify the absolute changes the variation in parameter values makes.
Response
Thanks for your advice. We have added the Table 6 in paper to describe the variation of pharmacokinetics parameters.
Point 27
Figure 4, it is difficult to see the difference between Whole and non-gut.
Response:
Thanks for your advice. It is really true as you suggested that it is hard to separate the “non-gut” group and “whole” group in the same figure. We have tried our best to make it distinguishable. And we also have added the Table 6 which describe the variation of pharmacokinetics parameters, and this may make it easier to identify them.
Point 28
Please refrain from using phrases like “most of”. Please use phrases that mention the actual percentage/ratio/increase.
Response
Thanks for your advice. We have changed the “most of” to the accurate percentage value as following:
“The results showed that 62% predicted concentrations (24/39) and 96% predicted exposure parameters (75/78) fell within 0.5-2.0-fold of observations, inferring that developed PBPK model was suitable for illustrating pharmacokinetic behaviors of CsA and the tested victim drugs.”
“96% of predications pharmacokinetics parameters fell within 0.5-2.0 folds of observation. Following validation, the developed PBPK model was successfully scaled to DDIs of CsA with 9 victim drugs. Except rosuvastatin, all DDI predictions had the same trend with observations, demonstrating that the CsA-mediated DDIs may be predicted using the developed PBPK model involving in enzymes and transporters in liver, intestine and kidney.”
Point 29
Page 11, line 336. As there was no table with the absolute values, it is difficult for the reader to verify this statement. But could it be that the effect of Ki,OATP was strongest because the parameter value was varied 5x more than the other parameters?
Response
Thanks for your advice. Sorry to confuse you. We have changed the contribution order based on the same multiplier.
CLint,uptake »SF>CsA dosage >Ki,OATP>Kt
The multiplier of factors such as Ki comes from the real value measured by the experiment [5,7], which had been discussed in Discussion section. So we didn’t add the 2-fold Ki,OATP in the manuscript.
Special thanks to you for your good comments.
Reference
- Mathialagan, S.; Piotrowski, M.A.; Tess, D.A.; Feng, B.; Litchfield, J.; Varma, M.V. Quantitative Prediction of Human Renal Clearance and Drug-Drug Interactions of Organic Anion Transporter Substrates Using In Vitro Transport Data: A Relative Activity Factor Approach. Drug Metab Dispos 2017, 45, 409-417, doi:10.1124/dmd.116.074294.
- Guo, H.; Liu, C.; Li, J.; Zhang, M.; Hu, M.; Xu, P.; Liu, L.; Liu, X. A mechanistic physiologically based pharmacokinetic-enzyme turnover model involving both intestine and liver to predict CYP3A induction-mediated drug-drug interactions. J Pharm Sci 2013, 102, 2819-2836, doi:10.1002/jps.23613.
- Wang, Z.; Yang, H.; Xu, J.; Zhao, K.; Chen, Y.; Liang, L.; Li, P.; Chen, N.; Geng, D.; Zhang, X., et al. Prediction of Atorvastatin Pharmacokinetics in High-Fat Diet and Low-Dose Streptozotocin-Induced Diabetic Rats Using a Semiphysiologically Based Pharmacokinetic Model Involving Both Enzymes and Transporters. Drug Metab Dispos 2019, 47, 1066-1079, doi:10.1124/dmd.118.085902.
- Kim, S.J.; Toshimoto, K.; Yao, Y.; Yoshikado, T.; Sugiyama, Y. Quantitative Analysis of Complex Drug-Drug Interactions Between Repaglinide and Cyclosporin A/Gemfibrozil Using Physiologically Based Pharmacokinetic Models With In Vitro Transporter/Enzyme Inhibition Data. J Pharm Sci 2017, 106, 2715-2726, doi:10.1016/j.xphs.2017.04.063.
- Varma, M.V.; Bi, Y.A.; Kimoto, E.; Lin, J. Quantitative prediction of transporter- and enzyme-mediated clinical drug-drug interactions of organic anion-transporting polypeptide 1B1 substrates using a mechanistic net-effect model. J Pharmacol Exp Ther 2014, 351, 214-223, doi:10.1124/jpet.114.215970.
- Varma, M.V.; Lai, Y.; Kimoto, E.; Goosen, T.C.; El-Kattan, A.F.; Kumar, V. Mechanistic modeling to predict the transporter- and enzyme-mediated drug-drug interactions of repaglinide. Pharm Res 2013, 30, 1188-1199, doi:10.1007/s11095-012-0956-5.
- Izumi, S.; Nozaki, Y.; Komori, T.; Maeda, K.; Takenaka, O.; Kusano, K.; Yoshimura, T.; Kusuhara, H.; Sugiyama, Y. Substrate-dependent inhibition of organic anion transporting polypeptide 1B1: comparative analysis with prototypical probe substrates estradiol-17beta-glucuronide, estrone-3-sulfate, and sulfobromophthalein. Drug Metab Dispos 2013, 41, 1859-1866, doi:10.1124/dmd.113.052290.

Reviewer 3 Report
Authors present PBPK models to predict disposition CsAs and 9 drugs which have DDI with CsAs in this manuscript. Here are some comments and suggestions.
Many reviews and individual manuscripts on PBPK models are published. Authors should clearly mention in the introduction, what this manuscript adds new.
Line 64 - Pharmacokinetics of cerivastatin was "calculated". Use "calculated" or some other word instead of "operated" through out the manuscript.
Line 38 - Although within range, predictions are not near clinical observations. Remove the claim - ' near the observations' .
Author Response
Dear Reviewer:
Thanks for your comments, which are all valuable and very helpful for revising and improving our paper, as well as the important guiding significance to our researches. We have studied comments carefully and have made correction which we hope meet with approval. Revised portion are marked in red in the paper. The main corrections in the paper and the responds to the reviewer’s comments are as flowing:
Point 1
Many reviews and individual manuscripts on PBPK models are published. Authors should clearly mention in the introduction, what this manuscript adds new.
Response:
Thanks for your advice. We have added the formulation in the section Introduction as following.
“There have been researches about the DDI between statin and CsA with PBPK model [1,2], but few have integrated simultaneously of liver, intestinal and kidney on drug disposal. The highlight of our work is to investigate the integrity effect of enzymes and transporters in liver, intestinal and kidney on drug disposition using a whole PBPK model characterizing interplay of enzyme-transporter in liver, intestine and kidney. The developed PBPK first was used to predict disposition of CsA and 9 victim drugs (OATP substrates) including atorvastatin, cerivastatin, pravastatin, rosuvastatin, fluvastatin, simvastatin, lovastatin, repaglinide and bosentan as well as DDIs of CsA with 9 victim drugs. The predictions were compared with clinic reports. Atorvastatin, substrate of uptake transporter (OATP), efflux transporter (BCRP, P-gp, MRP) and enzyme CYP3A, was represented to further investigate individual contributions of transporters and CYP3A4 to disposition of drugs and to document sensitivity analysis.”
Point 2
Line 64 - Pharmacokinetics of cerivastatin was "calculated". Use "calculated" or some other word instead of "operated" throughout the manuscript.
Response:
Thanks for your advice. The "operated" has been changed as “implemented”.
Point 3
Although within range, predictions are not near clinical observations. Remove the claim - ' near the observations'.
Response
Thanks for your advice. I am sorry not to clearly illustrate it. Fold error was made to be the criteria for prediction successful or not. The fold errors of both AUC and Cmax between observed and predicted data were less than 2, indicating validity of the PBPK model in simulations. We used these criteria in the whole paper. To make the work more precise, we have changed the statement as “within fold error range”.
Special thanks to you for your good comments.
Reference
- Varma, M.V.; Bi, Y.A.; Kimoto, E.; Lin, J. Quantitative prediction of transporter- and enzyme-mediated clinical drug-drug interactions of organic anion-transporting polypeptide 1B1 substrates using a mechanistic net-effect model. J Pharmacol Exp Ther 2014, 351, 214-223, doi:10.1124/jpet.114.215970.
- Varma, M.V.; Lai, Y.; Feng, B.; Litchfield, J.; Goosen, T.C.; Bergman, A. Physiologically based modeling of pravastatin transporter-mediated hepatobiliary disposition and drug-drug interactions. Pharm Res 2012, 29, 2860-2873, doi:10.1007/s11095-012-0792-7.

Reviewer 4 Report
The manuscript entitled "Prediction of cyclosporin-mediated drug interaction using physiologically based pharmacokinetic model characterizing interplay of drug transporters and enzymes." provides novel data on PBPK models of the interaction between cyclosporine and 9 interacting drugs.
I have the following comments:
1- Line 25 (and methods) Most of predictions were within 0.5-2.0 folds of observations. To present a 2-fold difference in PK parameters, do the authors consider it to be “suitable”? Could they provide a statistical measure of “being suitable” and the corresponding reference?.
2- It is unclear the aim of choosing the 9 victim drugs.
3- Line 26 Atorvastatin was represented to investigate individual contributions of transporters and CYP3As to atorvastatin disposition and their integrated effect. It is very confusing the abstract, as first it is stated that a PBPK model was constructed for cyclosporine and 9 victims and then, it refers only to atorvastatin and no mention to cyclosporine.
4- Could the authors make themselves more explicit the aim of the study? Why were these drugs chosen? what's the clinical implication? It is confusing the current version.
5-Why did the authors select atorvastatin to perform a separate study? It seems like there are several aims not well described but interconnected. Please, briefly explain in the introduction and provide adequate references.
6- From line 79 to 88 the authors present data that should be in the introduction and methods. Why is that information in results?
7- Figure 1 has no letters in the figure to identify the different panels.
8- I think that (as no letters are available, I am not sure which drug is) for simvastatin, the profile looks like the drug presents a lag time. Have the authors tried to change the parameters to consider such a feature?
9- These should be in methods: The DDIs were indexed as ratio of AUC or Cmax (AUCR) for victim drugs with CsA to without CsA. Dosage of CsA was set to be the highest dosage reported in clinic reports
10- The authors extensively revised two reports of interactions between atorvastatin and cyclosporine in the Results section. This is completely out of focus in this section. Reconsider to modify the modify the whole manuscript to place the correct information in the corresponding sections. The same happens with the other victim drugs. Then, it comes Figure 2 after a thorough description of the PK of each drug, not even being cited in the text close to that figure.
11- Could the authors provide references to the sensitivity analysis proposed by the them?
12- Check the grammar and misspelling all over the manuscript but pay attention specially to the following sentences:
The results got the conclusion that the developed PBPK model characterizing interplay of enzymes and transporters was successfully applied to predict pharmacokinetics of 10 OATP substrates and DDIs of CsA with 31 9 victim drugs.
2.1 Collection of drug and their parameters used for PBPK prediction
Physiological parameters of human such as organ volume, blood flow rate, metabolic and transport parameters of CsA and victim drugs were cited from publications PubMed.
The results demonstrated that no considering inhibitions of CsA on hepatic CYP3A-mediated metabolism and integrated effects occurring in liver or intestine led to remarkably lower concentrations of atorvastatin 285 compared with actual concentrations of atorvastatin when co-administrated CsA.
Atorvastatin was used for mode drug further to investigate individual contributions of 298 CYP3A, OATPs, P-gp, MRP and BCRP inhibitions to atorvastatin disposition (Figure 4).
A whole PBPK model charactering interplay of enzymes and transporters in intestine, liver and kidney (Figure 5) was developed to predict pharmacokinetic behaviors of victim drugs and CsA as well as DDIs of CsA with these victim drugs
Author Response
Dear Reviewer:
Thanks for your comments, which are all valuable and very helpful for revising and improving our paper, as well as the important guiding significance to our researches. We have studied comments carefully and have made correction which we hope meet with approval. Revised portion are marked in red in the paper. The main corrections in the paper and the responds to the reviewer’s comments are as flowing:
Point 1
Line 25 (and methods) Most of predictions were within 0.5-2.0 folds of observations. To present a 2-fold difference in PK parameters, do the authors consider it to be “suitable”? Could they provide a statistical measure of “being suitable” and the corresponding reference?
Response:
Thanks for your advice. We are sorry for unclear statement about this. Fold error was made to be the criteria for prediction successful or not. Fold errors of both AUC and Cmax between observed and predicted data were less than 2, indicating validity of the PBPK model in simulations. We have added this criterion in the test. There are many research use such fold error to be the criteria, and we provide two of which to make the statement stronger. [1,2]
Point 2:
It is unclear the aim of choosing the 9 victim drugs.
Response:
Thanks for your advice. Clinical DDI studies involving CsA and OATP substrates were collected from publications on Pubmed. CsA-mediated DDIs were based on the following criterions. 1) Data might come from healthy subjects or patients following single dose or multidose of victim drugs when co-administrated with CsA. 2) CsA was orally administrated to subjects in immediate release formulation. 3) Pharmacokinetic parameters such as peak concentration (Cmax) or area under the curve (AUC) or pharmacokinetic profiles were shown. 4) DDI data and the data for drugs used alone might come from different reports.
We looked through reports on PubMed, only 9 victims met the criteria.
Point 3
Line 26 Atorvastatin was represented to investigate individual contributions of transporters and CYP3As to atorvastatin disposition and their integrated effect. It is very confusing the abstract, as first it is stated that a PBPK model was constructed for cyclosporine and 9 victims and then, it refers only to atorvastatin and no mention to cyclosporine.
Response
Thanks for your advice. We are sorry to confuse you. Something we didn’t explain clear. There were 3 steps in the simulations. Firstly, simulations of the plasma concentration-time curves of the 9 victim drugs and CsA was performed. Secondly, the simulations of the plasma concentration-time curves of the 9 victim drugs coadministered with CsA were performed. Thirdly, set atorvastatin as representative one to operate sensitivity analyzation and investigate contribution of enzymes and transporters. We have added this in Method as following:
“There were two steps in this process. Pharmacokinetic profiles and their parameters (AUC and Cmax) of 9 victim drugs and CsA following oral administration to human were first predicted using the developed PBPK models and compared with clinic observations. Following validation, the developed PBPK model was separately used to predict DDIs of CsA with victim drugs and confirmed with clinic data. Extent of DDI was assessed as ratio (AUCR) of AUC or (CmaxR) of Cmax with CsA and without CsA. Dosage of CsA was set to be the highest dosage reported in clinic reports. The predictions were considered successful if the ratio of predication to observation fell between 0.5 and 2.0.”
Point 4
Could the authors make themselves more explicit the aim of the study? Why were these drugs chosen? what's the clinical implication? It is confusing the current version.
Response:
Thanks for your advice. We are very sorry for our unclear writing. The idea for this experiment is to confirm that drug disposition is the cooperation of various transporter and metabolic enzymes. The whole-body PBPK model as an auxiliary tool to make the fact clearer and visualization. Moreover, from this way we also can exposed the different contributions of this transporters and enzymes. We want to make good use of this tool (PBPK model) to explain the phenomenon occurs during drug disposition. From this idea, with the previous work in the comprehensive laboratory [3], we firstly identified two drugs, atorvastatin and cyclosporin, and operate present research. Then, we find other drugs which are substance of transporter OATP following the response for point 2. The drugs all have DDI potential with cyclosporin. We have rewritten this part in Introduction section as following:
“There have been researches about the DDI between statin and CsA with PBPK model [27,28]. But few have integrated simultaneously of liver, intestinal and kidney on drug disposal. The highlight of our work is to investigate the integrity effect of enzymes and transporters in liver, intestinal and kidney on drug disposition using PBPK model as a powerful tool. We have established a whole-body PBPK model which predict disposition of CsA and 9 victim drugs (OATP substrates) we selected including atorvastatin, cerivastatin, pravastatin, rosuvastatin, fluvastatin, simvastatin, lovastatin, repaglinide and bosentan as well as DDIs of CsA with 9 victim drugs to investigate the integrity effect of enzymes and transporters in liver, intestinal and kidney on drug disposition. The predictions were compared with clinic reports. Atorvastatin, substance of uptake transporter (OATP), efflux transporter (BCRP, P-gp, MRP) and enzyme CYP3A, which were the focus of our research was represented to further investigate individual contributions of transporters and CYP3A4 to disposition of drugs and to document sensitivity analysis.”
Point 5
Why did the authors select atorvastatin to perform a separate study? It seems like there are several aims not well described but interconnected. Please, briefly explain in the introduction and provide adequate references.
Response:
Thanks for your advice. We have added these statements to Introduction section following your suggestion. For the reason to select atorvastatin as a typical drug: (1) atorvastatin was substance of uptake transporter (OATP), efflux transporter (BCRP, P-gp, MRP) and enzyme CYP3A, which were the focus of our research; (2) we wanted to reveal the interplay-effect of these transporters and enzymes in liver, kidneys and intestinal on drug disposition, which never been reported before, and atorvastatin met our requirements very well; (3) previous work in the laboratory had done about atorvastatin in rat.
“The predictions were compared with clinic reports. Atorvastatin, substrate of uptake transporter (OATP), efflux transporter (BCRP, P-gp, MRP) and enzyme CYP3A, was represented to further investigate individual contributions of transporters and CYP3A4 to disposition of drugs and to document sensitivity analysis.”
Point 6
From line 79 to 88 the authors present data that should be in the introduction and methods. Why is that information in results?
Response:
Thanks for your advice. Sorry to confuse you. Parameters in Table 1 were from reference. Parameters in Table 2 were from reference or calculated based on reference. Parameters in Table 3 were from reference or estimated. These data were all the basic of PBPK model we built. We tried to remove the Table 1-3, but the journal has the rule with “Figures should be placed in the main text near to the first time they are cited”. And in result section, the data used to estimate had to mentioned. So these data were in Method section.
Point 7
Figure 1 has no letters in the figure to identify the different panels.
Response:
Thanks for your advice. We are very sorry for our negligence and we have added the letters.
Point 8: I think that (as no letters are available, I am not sure which drug is) for simvastatin, the profile looks like the drug presents a lag time. Have the authors tried to change the parameters to consider such a feature?
Response:
Thanks for your advice. Following your suggestion, we make specify in the Results section. 6-membered lactone ring of simvastatin is hydrolyzed in vivo to generate the beta, delta-dihydroxy acid, an active metabolite structurally similar to HMG-CoA (hydroxymethylglutaryl CoA). Simvastatin is a prodrug. While the active metabolite simvastatin acid is substance of the transporters and enzymes mentioned before. For this reason, we took the observed concentration of simvastatin acid as reference. The lag time can be interpreted as the time for simvastatin (lactone) to simvastatin acid.
“Special version for simvastatin. 6-membered lactone ring of simvastatin is hydrolyzed in vivo to generate the beta, delta-dihydroxy acid, an active metabolite structurally similar to HMG-CoA (hydroxymethylglutaryl CoA). While the active metabolite simvastatin acid is substance of the transporters and enzymes mentioned before. We took the observed concentration of simvastatin acid as reference. The lag time of observations can be interpreted as the time for simvastatin (lactone) to simvastatin acid.”
Point 9
These should be in methods: The DDIs were indexed as ratio of AUC or Cmax (AUCR) for victim drugs with CsA to without CsA. Dosage of CsA was set to be the highest dosage reported in clinic reports.
Response:
Thanks for your advice. We have added these statements to Method section.
Point 10
The authors extensively revised two reports of interactions between atorvastatin and cyclosporine in the Results section. This is completely out of focus in this section. Reconsider to modify the whole manuscript to place the correct information in the corresponding sections. The same happens with the other victim drugs. Then, it comes Figure 2 after a thorough description of the PK of each drug, not even being cited in the text close to that figure.
Response:
Thanks for your advice. We are sorry not to clearly illustrated. DDI of cyclosporin A with atorvastatin was also our focus. We have revised the statement in Result section and changed the position of Figure 2.
“The Pharmacokinetic profiles for coadministration of perpetrator CsA and victim drugs (atorvastatin, cerivastatin, pravastatin, rosuvastatin, fluvastatin, simvastatin, lovastatin, repaglinide and bosentan) were showed in Figure 2 and Table 5. The detail interpretation will introduce one by one for pharmacokinetic of each drug administrated with and without cyclosporine.”
Point 11
Could the authors provide references to the sensitivity analysis proposed by the them?
Response
Thanks for your advice. We have added the reference in sensitivity analysis [3]. More detail information about sensitivity analysis has been mentioned in Discussion section.
Point 12
Check the grammar and misspelling all over the manuscript.
Response
Thanks for your advice. We have re-examined and revised the English expression of the manuscript.
Special thanks to you for your good comments.
Reference
- Liu, X.I.; Momper, J.D.; Rakhmanina, N.; van den Anker, J.N.; Green, D.J.; Burckart, G.J.; Best, B.M.; Mirochnick, M.; Capparelli, E.V.; Dallmann, A. Physiologically Based Pharmacokinetic Models to Predict Maternal Pharmacokinetics and Fetal Exposure to Emtricitabine and Acyclovir. J Clin Pharmacol 2020, 60, 240-255, doi:10.1002/jcph.1515.
- Parrott, N.; Paquereau, N.; Coassolo, P.; Lave, T. An evaluation of the utility of physiologically based models of pharmacokinetics in early drug discovery. J Pharm Sci 2005, 94, 2327-2343, doi:10.1002/jps.20419.
- Wang, Z.; Yang, H.; Xu, J.; Zhao, K.; Chen, Y.; Liang, L.; Li, P.; Chen, N.; Geng, D.; Zhang, X., et al. Prediction of Atorvastatin Pharmacokinetics in High-Fat Diet and Low-Dose Streptozotocin-Induced Diabetic Rats Using a Semiphysiologically Based Pharmacokinetic Model Involving Both Enzymes and Transporters. Drug Metab Dispos 2019, 47, 1066-1079, doi:10.1124/dmd.118.085902.
Round 2
Reviewer 1 Report
Even though the authors described and answered this reviewer’s comments, there are still not enough answers (or explanations) and the revised manuscript still remains questionable in major parts. The presented PBPK model simply incorporated the interplay between enzymes and transporters in intestine only. Moreover, they constructed the PBPK model simply based on the intestinal segments such as duodenum, jejunum and ileum, not based on the main intestinal segment or enterocytes in which the interplay between enzymes and transports actually occurs. This is the main concern of the presented PBPK model. Therefore, each intestinal transit time is very important to this interplay. Thus, the calculated or estimated transit times are substantially key parameters for the PBPK model. However, the 31 hr used in this PBPK model is still too high and unrealistic. The references [3, 5] suggested by authors also used realistic intestinal transit time within 24 hr. If the authors applies shorter transit time to the PBPK model, the CmaxR and absolute average fold error (AAFE) which the authors suggested the main parameters in this manuscript will be changed. This is the second concern. On the other hand, among 9 victim drugs, biliary and/or renal excretion play very important roles in their disposition in the body. Also, the ratio of drug concentration in tissue to plasma is very important factor for the PBPK model. The relative magnitude of this ratio to the organ blood flow is the key factor for the development and construction of PBPK model. Think about this relative magnitude in liver and kidney with high ratio (Kp) victim drugs based on the presented PBPK model. Thus, the PBPK model dose not fully reflect the disposition characteristics of 9 victim drugs and neither lead to the proper interpretation for DDI with CsA. These comments do not simply mean incorporating each victim compound’s characteristics in the discussion section. The authors would be better to select or focus the victim drugs among 9 victim drugs to be able to interpret the DDI with CsA using the presented PBPK model. This is the third concern.Author Response
Dear Reviewer:
Thanks for your comments, which are all valuable and very helpful for revising and improving our paper, as well as the important guiding significance to our researches. We have studied comments carefully and have made correction which we hope meet with approval. Revised portion are marked in red in the paper. The main corrections in the paper and the responds to the reviewer’s comments are as flowing:
Point 1:
The presented PBPK model simply incorporated the interplay between enzymes and transporters in intestine only. Moreover, they constructed the PBPK model simply based on the intestinal segments such as duodenum, jejunum and ileum, not based on the main intestinal segment or enterocytes in which the interplay between enzymes and transports actually occurs. This is the main concern of the presented PBPK model.
Response 1
We are sorry not to clearly it. We have divided the Table 3 to Table 3 and Table 4 to make it easy to identify showing following and the manuscript. The developed PBPK clearly exhibited the interplay between enzymes and transporters which occurs in liver, intestinal and kidney.
In liver:
CLint,uptake, CLint,bile and CLint,met represented uptake of hepatocyte from blood by OATP, efflux of hepatocyte to bile by BCRP/P-gp/MRP and metabolism by CYP3A in liver, respectively, involving in interplay of OATP, CYP3A4 and BCRP/P-gp/MRP2 in liver. The equation 10 in the method illustrated the interplay of transporters and enzymes in liver.
|
|
(10) |
where CLint, up, OATP and CLint, pd were OATP-mediated intrinsic clearance of uptake and passive clearance, respectively. CLint, met, CYP3A and CLint, met, non-CYP3A were intrinsic clearances of CYP3A-mediated metabolism and non-CYP3A-mediated metabolism, respectively. CLint, bile was biliary excretion clearance by P-gp, BCRP or MRP2. Ki, OATP, Ki, CYP3A and Ki, P-gp/BCRP/MRP2 were inhibition constants of OATP-mediated uptake, CYP3A-mediated metabolism and P-gp, BCRP-or MRP2-mediated biliary excretion by CsA, respectively.”
Table 3 showed metabolic/transport intrinsic clearances of the tested drugs. The results showed that among the tested drugs, eight drugs (cyclosporin A, atorvastatin, cerivastatin, fluvastatin, simvastatin, lovastatin, repaglinide and bosentan) are involved in interplay of OATPs and CYP3A. two drugs (pravastatin and rosuvastatin) are involved in interplay of OATPs and P-gp or BCRP. And 7 drugs (cyclosporin A, atorvastatin, cerivastatin, fluvastatin, simvastatin, repaglinide and bosentan) are involved in interplay of OATPs, CYP3A and P-gp, BCRP or MRP2.
Table 3. Intrinsic metabolic/transport parameters and empirical scaling factor (SF) of cyclosporin A (CsA), atorvastatin (Ato), cerivastatin (Cer), pravastatin (Pra), rosuvastatin (Ros), fluvastatin (Flu), simvastatin (Sim), lovastatin (Lov), repaglinide (Rep) and bosentan (Bos) in liver.
|
Drug |
CLint met |
CLint uptake |
SF |
CLint, pd |
CLint, bile |
|||
|
CYP3A4 |
other |
P-gp |
BCRP |
MRP2 |
||||
|
mL/min |
mL/min |
mL/min |
mL/min |
|||||
|
CsA |
5432 [1] |
/ |
10857 [1] |
0.1 a |
2933 [1] |
637 [1] |
/ |
/ |
|
Ato |
3469.5 [2] |
612.3 [2] |
6374.7 [3] |
32.6 [2] |
3916.79 [1] |
/ |
302.4 [4] |
/ |
|
Cer |
979.8 [2] |
1197.6 [2] |
1942.8 [5] |
12.5 [2] |
3541.5 [2] |
40.5 [2] |
/ |
/ |
|
Pra |
/ |
/ |
283.3 [2] |
19.4 [2] |
80.9 [2] |
/ |
/ |
80.9 [2] |
|
Ros |
/ |
/ |
1841.6 [2] |
9.2 [2] |
242.8 [2] |
/ |
566.6 [2] |
/ |
|
Flu |
2386.3 [5] |
2801.3 [5] |
9106.6 [5] |
21 a |
4047.4 [5] |
/ |
3440 [5] |
/ |
|
Sim |
231391 [6] |
/ |
9234.1 [7] |
25 a |
23695.5 [7] |
101.2 [7] |
/ |
/ |
|
Lov |
6893.96 [8] |
/ |
13129.8 [7] |
4 a |
11569.5 [7] |
/ |
/ |
/ |
|
Rep |
2503.7 [2] |
6438 [2] |
7184 [2] |
16.9 [9] |
4452.14 [2] |
/ |
20.24 [2] |
/ |
|
Bos |
1365.14 [2] |
/ |
7184 [2] |
1.1 [2] |
2023.7 [2] |
1472.6 [10] |
/ |
/ |
a SF is active uptake scaling factor from the simulation which source has been shown in Table S1 in Supplementary Material.
In intestine.
Duodenum, jejunum and ileum were also individually considered. Several reports about the whole-body PBPK model also used the same intestinal segments [11-13]. The transport parameters mediated by BCRP/P-gp/MRP2 and CYP3A-mediated metabolism parameters in each intestinal segments were individually estimated according to their expression. Constants of transit rate in each segment was also introduced in the PBPK model. The parameters for intestinal to build the PBPK model were shown in Table 4.
Table 4. Intrinsic metabolic/transport parameters of cyclosporin A (CsA), atorvastatin (Ato), cerivastatin (Cer), pravastatin (Pra), rosuvastatin (Ros), fluvastatin (Flu), simvastatin (Sim), lovastatin (Lov), repaglinide (Rep) and bosentan (Bos) in intestine.
|
Drug |
ka |
Peff, A-B |
Peff, B-A |
CLint, gut [13] |
|||||||||||
|
|
min-1 |
cm/min |
|
cm/min |
|
|
mL/min |
|
|||||||
|
|
P-gp |
BCRP |
MRP2 |
Duodenum |
Jejunum |
Ileum |
|||||||||
|
CsA |
0.025 [14] |
/ |
0.01 [14] |
/ |
/ |
11.25 |
44.6 |
25.98 |
|||||||
|
Ato |
/ |
0.0094 [15] |
0.0185 [16] |
0.0118 [16] |
0.011 [16] |
12.2 |
48.273 |
28.22 |
|||||||
|
Cer |
/ |
0.0133 [17] |
0.0099 [18] |
0.0065 [18] |
0.0118 [18] |
/ |
/ |
/ |
|||||||
|
Pra |
0.021 [14] |
/ |
/ |
/ |
43.7 a |
/ |
/ |
/ |
|||||||
|
Ros |
0.0022 [19] |
/ |
0.012 [16] |
0.013 [16] |
0.011 [16] |
/ |
/ |
/ |
|||||||
|
Flu |
/ |
0.038 [20] |
0.035 [16] |
0.026 [16] |
0.029 [16] |
/ |
/ |
/ |
|||||||
|
Sim |
/ |
0.01 [21] |
0.0093 [21] |
/ |
/ |
755.2 |
2991.3 |
1745 |
|||||||
|
Lov |
/ |
0.016 [21] |
/ |
0.014 [21] |
871 |
89.8 |
2011 |
||||||||
|
Rep |
0.04 [9] |
0.0148 [22] |
/ |
/ |
/ |
||||||||||
|
Bos |
/ |
0.014 [10] |
|
5.84 b [10] |
/ |
/ |
/ |
||||||||
a The unit of value is mL/min for the efflux of MRP2.
b The value of Vmax with unit pmol/min and Km with unit μM is stand for the efflux effect of MRP2.
The equation 6 in the manuscript also reflected interplay of transporters (P-gp, BCRP or MRP2) and CYP3A in intestinal segments.
|
|
Where Kti, ka,i and Agwi were gut transit rate constant, absorption rate constant in the ith gut segment and the drug amount in the ith enterocyte compartment. kb,i was efflux rate constant of drug from enterocytes into lumen, which was attributed to intestinal efflux transporters(P-gp, BCRP or MRP2). The kbi was estimated using corrected Peff,man B-A by transporter expression in the ith segments.
CLint,gut was intrinsic clearance of CYP3A mediated metabolism for drugs in the ith intestinal segment, which was estimated using CYP3A4 expression or activity in corresponding segment.
The intestinal efflux of the tested 10 drugs was involved in efflux transporters (P-gp, BCRP or MRP2). Intestinal CYP3A mediated metabolism of atorvastatin, CsA, simvastatin and lovastatin.
In renal:
Renal secretions clearance of rosuvastatin and pravastatin were mediated by renal BCRP and MRP2, respectively.
We have added the description of CLren in Method section and we also mentioned this in Result section showing blow.
In Method
CLint, GFR of pravastatin and rosuvastatin were 132.9 mL/min. CLint,sec of pravastatin (MRP2) and rosuvastatin (BCRP) were 1538.96 mL/min and 420.3 mL/min, respectively.
In Results
“Renal efflux transporters (BCRP, P-gp and MRP2) also mediated renal excretion of atorvastatin, although its contribution to total clearance was minor.”
“Pravastatin is a substrate of MRP2 and OATPs. The drug mainly eliminates via MRP2-mediated biliary and MRP2-mediated renal secretion, indicating that CsA leads to DDI with pravastatin via affecting OATP-mediated hepatic uptake, MRP2-mediated intestinal efflux, MRP2-mediated biliary excretion and MRP2-mediated renal secretion.”
“Rosuvastatin is a substrate of P-gp, BCRP, MRP2 and OATPs. Rosuvastatin mainly eliminates via BCRP-dependent biliary and renal secretion.”
“Fluvastatin is metabolized by CYP2C9. Fluvastatin is also substrate of P-gp, BCRP, MRP2 and OATPs. Liver BCRP also mediates biliary excretion of fluvastatin. Thus, DDI of CsA with fluvastatin mainly results from inhibiting above transporters.”
“Bosentan mainly eliminates via metabolism mechanism, which partly is attributed to CYP3A4. Bosentan is also a substrate of MRP2 and OATPs and renal MRP2 may be involved in its renal excretion of bosentan. These results inferred that DDI of CsA with bosentan is attributed to inhibition of CYP3A4-mediated metabolism, OATPs-mediated hepatic uptake and MRP2-mediated renal secretion.”
Point 2:
Therefore, each intestinal transit time is very important to this interplay. Thus, the calculated or estimated transit times are substantially key parameters for the PBPK model. However, the 31 hr used in this PBPK model is still too high and unrealistic. The references [3,5] suggested by authors also used realistic intestinal transit time within 24 hr. If the authors applies shorter transit time to the PBPK model, the CmaxR and absolute average fold error (AAFE) which the authors suggested the main parameters in this manuscript will be changed.
Response 2:
Thanks for your advice. We have changed the Kt of cecum and colon from 0.0025 and 0.00085 to 0.004 and 0.0013, respectively. The intestinal transit time are within 24 hr after the modification. It was generally accepted that cecum and colon little contributed to drug absorption, transit time in cecum and colon did not affect plasma concentrations of drug. Following table compared Cmax and AUC of atorvastatin following 40 mg using different constant of transit rate.
Table 1. Cmax and AUC of atorvastatin using different constant of transit rate.
|
Kt for cecum and colon |
Dose |
Cmax |
AUC |
|
min -1 |
mg |
ng/ml |
ng·h/mL |
|
0.0025 and 0.00085 |
40 |
14.59 |
71.69 |
|
0.004 and 0.0013 |
40 |
14.59 |
71.69 |
Point 3:
On the other hand, among 9 victim drugs, biliary and/or renal excretion play very important roles in their disposition in the body. Also, the ratio of drug concentration in tissue to plasma is very important factor for the PBPK model. The relative magnitude of this ratio to the organ blood flow is the key factor for the development and construction of PBPK model. Think about this relative magnitude in liver and kidney with high ratio (Kp) victim drugs based on the presented PBPK model. Thus, the PBPK model dose not fully reflect the disposition characteristics of 9 victim drugs and neither lead to the proper interpretation for DDI with CsA. These comments do not simply mean incorporating each victim compound’s characteristics in the discussion section. The authors would be better to select or focus the victim drugs among 9 victim drugs to be able to interpret the DDI with CsA using the presented PBPK model.
Response:
Thanks for your advice. We are very sorry for our obscure expression. The test drugs are all the substrates of OATPs. High Kp values of drugs in liver were mainly attributed to OATP-mediated hepatic uptake of drugs. The elimination of drugs in the liver may be simply illustrated using following Figure I.
Figure I. Schematic model representing the elimination of drugs in the liver. Drugs are taken up into the liver across the sinusoidal membrane. Then some molecules are metabolized, or excreted into bile or exported to blood. Symbol: CLint,all, overall hepatic intrinsic clearance; CLint,up, intrinsic uptake clearance; CLint,back, intrinsic clearance of backflux to blood; CLint,met, intrinsic metabolism clearances; CLint,bile, biliary clearance of unbound drug; QH, hepatic blood flow. Cin and Cout concentrations of drug in arterial and venous blood, respectively.
The overall hepatic intrinsic clearance (CLint,all) is expressed as a hybrid parameter consisting of membrane permeation clearances of unbound parent drugs for influx (CLint,up) and efflux (CLint,back) across the sinusoidal membrane, the intrinsic clearance for metabolism (CLint,met) and biliary excretion of unbound drugs(CLint,bile), i.e
If CLint,bile +CLint,met >>CLint,back,
CLint,all =CLint,up.
For substrate of OATPs
CLint, up=CLint,OATPs+CLint, passive
In general, CLint,OATPs>CLint, passive
Thus, CLint,all »CLint, up»CLint,OATPs
This is to say that overall hepatic intrinsic clearance can be described by OATP-mediated hepatic uptake.
CsA strongly inhibited OATP-mediated hepatic uptake, leading to distribution of drugs (Table 2), decreasing hepatic clearances of drug and increasing plasma exposure of drugs, leading to DDI.
Table 2. The uptake clearance by OATP of atorvastatin used alone and co-administrated with CsA.
|
Sort |
CLOATP |
|
|
mL/min |
|
Alone |
6374.7 |
|
DDI |
925.6 |
We also added this in Introduction section as following.
“Hepatic uptake of these drugs is mainly mediated by hepatic OATPs, which lead to high Kp values of drugs, metabolism of which is by CYP450s (main CYP3A4) and biliary or renal excretion of which is via a P-gp, BCRP or MRP2-dependent mechanism.”
Special thanks to you for your good comments again!
- Camenisch, G.; Umehara, K. Predicting human hepatic clearance from in vitro drug metabolism and transport data: a scientific and pharmaceutical perspective for assessing drug-drug interactions. Biopharm Drug Dispos 2012, 33, 179-194, doi:10.1002/bdd.1784.
- Varma, M.V.; Bi, Y.A.; Kimoto, E.; Lin, J. Quantitative prediction of transporter- and enzyme-mediated clinical drug-drug interactions of organic anion-transporting polypeptide 1B1 substrates using a mechanistic net-effect model. J Pharmacol Exp Ther 2014, 351, 214-223, doi:10.1124/jpet.114.215970.
- Morse, B.L.; Alberts, J.J.; Posada, M.M.; Rehmel, J.; Kolur, A.; Tham, L.S.; Loghin, C.; Hillgren, K.M.; Hall, S.D.; Dickinson, G.L. Physiologically-Based Pharmacokinetic Modeling of Atorvastatin Incorporating Delayed Gastric Emptying and Acid-to-Lactone Conversion. CPT Pharmacometrics Syst Pharmacol 2019, 8, 664-675, doi:10.1002/psp4.12447.
- Duan, P.; Zhao, P.; Zhang, L. Physiologically Based Pharmacokinetic (PBPK) Modeling of Pitavastatin and Atorvastatin to Predict Drug-Drug Interactions (DDIs). Eur J Drug Metab Pharmacokinet 2017, 42, 689-705, doi:10.1007/s13318-016-0383-9.
- Jones, H.M.; Barton, H.A.; Lai, Y.; Bi, Y.A.; Kimoto, E.; Kempshall, S.; Tate, S.C.; El-Kattan, A.; Houston, J.B.; Galetin, A., et al. Mechanistic pharmacokinetic modeling for the prediction of transporter-mediated disposition in humans from sandwich culture human hepatocyte data. Drug Metab Dispos 2012, 40, 1007-1017, doi:10.1124/dmd.111.042994.
- Nishimuta, H.; Sato, K.; Yabuki, M.; Komuro, S. Prediction of the intestinal first-pass metabolism of CYP3A and UGT substrates in humans from in vitro data. Drug Metab Pharmacokinet 2011, 26, 592-601, doi:10.2133/dmpk.DMPK-11-RG-034.
- Kunze, A.; Poller, B.; Huwyler, J.; Camenisch, G. Application of the extended clearance concept classification system (ECCCS) to predict the victim drug-drug interaction potential of statins. Drug Metab Pers Ther 2015, 30, 175-188, doi:10.1515/dmdi-2015-0003.
- Liu, Y.; Zeng, B.H.; Shang, H.T.; Cen, Y.Y.; Wei, H. Bama miniature pigs (Sus scrofa domestica) as a model for drug evaluation for humans: comparison of in vitro metabolism and in vivo pharmacokinetics of lovastatin. Comp Med 2008, 58, 580-587.
- Varma, M.V.; Lai, Y.; Kimoto, E.; Goosen, T.C.; El-Kattan, A.F.; Kumar, V. Mechanistic modeling to predict the transporter- and enzyme-mediated drug-drug interactions of repaglinide. Pharm Res 2013, 30, 1188-1199, doi:10.1007/s11095-012-0956-5.
- Sato, M.; Toshimoto, K.; Tomaru, A.; Yoshikado, T.; Tanaka, Y.; Hisaka, A.; Lee, W.; Sugiyama, Y. Physiologically Based Pharmacokinetic Modeling of Bosentan Identifies the Saturable Hepatic Uptake As a Major Contributor to Its Nonlinear Pharmacokinetics. Drug Metab Dispos 2018, 46, 740-748, doi:10.1124/dmd.117.078972.
- Qian, C.Q.; Zhao, K.J.; Chen, Y.; Liu, L.; Liu, X.D. Simultaneously predict pharmacokinetic interaction of rifampicin with oral versus intravenous substrates of cytochrome P450 3A/Pglycoprotein to healthy human using a semi-physiologically based pharmacokinetic model involving both enzyme and transporter turnover. Eur J Pharm Sci 2019, 134, 194-204, doi:10.1016/j.ejps.2019.04.026.
- Badhan, R.; Penny, J.; Galetin, A.; Houston, J.B. Methodology for development of a physiological model incorporating CYP3A and P-glycoprotein for the prediction of intestinal drug absorption. J Pharm Sci 2009, 98, 2180-2197, doi:10.1002/jps.21572.
- Gertz, M.; Houston, J.B.; Galetin, A. Physiologically based pharmacokinetic modeling of intestinal first-pass metabolism of CYP3A substrates with high intestinal extraction. Drug Metab Dispos 2011, 39, 1633-1642, doi:10.1124/dmd.111.039248.
- Varma, M.V.; Lai, Y.; Feng, B.; Litchfield, J.; Goosen, T.C.; Bergman, A. Physiologically based modeling of pravastatin transporter-mediated hepatobiliary disposition and drug-drug interactions. Pharm Res 2012, 29, 2860-2873, doi:10.1007/s11095-012-0792-7.
- Wu, X.; Whitfield, L.R.; Stewart, B.H. Atorvastatin transport in the Caco-2 cell model: contributions of P-glycoprotein and the proton-monocarboxylic acid co-transporter. Pharm Res 2000, 17, 209-215, doi:10.1023/a:1007525616017.
- Li, J.; Volpe, D.A.; Wang, Y.; Zhang, W.; Bode, C.; Owen, A.; Hidalgo, I.J. Use of transporter knockdown Caco-2 cells to investigate the in vitro efflux of statin drugs. Drug Metab Dispos 2011, 39, 1196-1202, doi:10.1124/dmd.111.038075.
- Varma, M.V.; Lin, J.; Bi, Y.A.; Kimoto, E.; Rodrigues, A.D. Quantitative Rationalization of Gemfibrozil Drug Interactions: Consideration of Transporters-Enzyme Interplay and the Role of Circulating Metabolite Gemfibrozil 1-O-beta-Glucuronide. Drug Metab Dispos 2015, 43, 1108-1118, doi:10.1124/dmd.115.064303.
- Kivisto, K.T.; Zukunft, J.; Hofmann, U.; Niemi, M.; Rekersbrink, S.; Schneider, S.; Luippold, G.; Schwab, M.; Eichelbaum, M.; Fromm, M.F. Characterisation of cerivastatin as a P-glycoprotein substrate: studies in P-glycoprotein-expressing cell monolayers and mdr1a/b knock-out mice. Naunyn Schmiedebergs Arch Pharmacol 2004, 370, 124-130, doi:10.1007/s00210-004-0948-z.
- Li, J.; Wang, Y.; Zhang, W.; Huang, Y.; Hein, K.; Hidalgo, I.J. The role of a basolateral transporter in rosuvastatin transport and its interplay with apical breast cancer resistance protein in polarized cell monolayer systems. Drug Metab Dispos 2012, 40, 2102-2108, doi:10.1124/dmd.112.045666.
- Winiwarter, S.; Bonham, N.M.; Ax, F.; Hallberg, A.; Lennernas, H.; Karlen, A. Correlation of human jejunal permeability (in vivo) of drugs with experimentally and theoretically derived parameters. A multivariate data analysis approach. J Med Chem 1998, 41, 4939-4949, doi:10.1021/jm9810102.
- Gertz, M.; Harrison, A.; Houston, J.B.; Galetin, A. Prediction of human intestinal first-pass metabolism of 25 CYP3A substrates from in vitro clearance and permeability data. Drug Metab Dispos 2010, 38, 1147-1158, doi:10.1124/dmd.110.032649.
- Yaghoobian, M.; Haeri, A.; Bolourchian, N.; Shahhosseini, S.; Dadashzadeh, S. An Investigation into the Role of P-Glycoprotein in the Intestinal Absorption of Repaglinide: Assessed by Everted Gut Sac and Caco-2 Cell Line. Iran J Pharm Res 2019, 18, 102-110.
Reviewer 2 Report
Overall
I thank the authors for their extensive and thorough answers to my questions and remarks. The authors have indeed improved their manuscript very much. I appreciate the time and effort the authors have put into their responses.
I have some smaller comments still left. Main comments are about having some additional data published in a supplementary. I suggest that these are clarified before the paper can be published.
Please see my remarks and questions from the first round in italics, and my current remarks and questions in bold.
M&M
Page 12, line 382-383: “4) DDI data might come from different reports.” What does this mean? Which DDI data does this regard? Does it mean that the PK data of the victim drug can come from one report and then the PK data of the victim drug coadministered with CsA comes from another report?
Thank you for the answer to this question. I think however, that it is still a bit unclear. I feel this point 4) is not really part of the selection criteria. Perhaps it could be a sentence behind the list?
Page 15, lines 458-466: How can the GFR be estimated (eq 14) if it was set to 120 ml/min/70kg?
Thank you for the answers to above question. Throughout this section on renal excretion, it is written that “XX might be calculated using equation…”. Was there any other way to calculate, or were all calculated with this equations? It is confusing when “might be” is introduced, because it suggests another method to get that parameter value.
Which parameters are estimated? To which data was the model fitted to estimate these parameters? What was the result of these estimations? All these questions should be addressed in the method section, and the results of the estimations should be presented clearly of course including the variability of the estimation.
Thank you for the elaborate answer to this question. It is easy to use the word estimate in the wrong instance. It is however extremely important to be aware of this problem, because it will create uncertainty about how the project was performed.
If the “SF values of cyclosporine, pravastatin, fluvastatin, simvastatin and lovastatin were estimated from the software based on their pharmacokinetic data”, could you provide the data on the curve fittings and parameter estimations and used observed data in a supplementary information? This way, the uncertainty in the curve fittings could be evaluated.
Why was there a difference in how much the parameters were changed (Ki,OATP was different)
Thank you for the answers to this question. I understand that it is interesting to evaluate the observed difference from literature. However, whilst Ki,OATP was changed according to observed, the other parameters were not treated the same way. This gives an unfair comparison to the other changed parameters. Either change all parameters with the observed variability and evaluate how much this might affect the AUC and Cmax. OR change all parameters with the same –fold, to have a fair comparison.
Results
To improve readers’ understanding, could the following sentence “was used to assess model performance in this part, as well as pharmacokinetics parameters, Cmaxand AUC.” be changed to “Model performance was assessed using absolute average fold error (AAFE) around each data point, as well as the ratio between observed and predicted Cmax and AUC”?
Thank you for including a new Table and updating Figure 2 to hours. That is really appreciated.
Is it possible to add the rest of the Pred/obs figures as a supplementary materials, if it makes the figures in the result section to “messy”?
Discussion
Page 11, line 336. As there was no table with the absolute values. But could it be that the effect of Ki,OATP was strongest because the parameter value was varied 5x more than the other parameters?
Thank you for clarifying your results. It could this data be added to the manuscript, so the readers understand why this is the correct contribution order? Otherwise, the discussion is based upon results not shown.
Author Response
Dear Reviewer:
Thanks for your comments, which are all valuable and very helpful for revising and improving our paper, as well as the important guiding significance to our researches. We have studied comments carefully and have made correction which we hope meet with approval. Revised portion are marked in red in the paper. The main corrections in the paper and the responds to the reviewer’s comments are as flowing:
Point 1: I feel this point 4) is not really part of the selection criteria. Perhaps it could be a sentence behind the list?
Response:
Thanks for your advice. It’s a good proposal. We have merged point 4 to point 1. Point 1 was also the criterion for the source of observed data.
“1) Data might come from healthy subjects or patients following single dose or multidose of victim drugs when co-administrated with CsA. What’s more, DDI data and the data for drugs used alone might come from different reports.”
Point 2: Throughout this section on renal excretion, it is written that “XX might be calculated using equation…”. Was there any other way to calculate, or were all calculated with this equations? It is confusing when “might be” is introduced, because it suggests another method to get that parameter value.
Response:
Thanks for your advice. We are very sorry for our incorrect writing. The equations was used to calculate renal excretion of all these drugs. On the other hand, the equation was the only one we used for all. We have corrected the vague expression.
“CLsec was calculated using the equation…”
“The CLint,GFR and CLint, sec were calculated using equation…”
Point 3: It is easy to use the word estimate in the wrong instance. It is however extremely important to be aware of this problem, because it will create uncertainty about how the project was performed.
Response:
Thanks for your advice. We agree with you. We will also pay great attention to the words we use in the future. Thank you very much for your reminding.
Point 4: If the “SF values of cyclosporine, pravastatin, fluvastatin, simvastatin and lovastatin were estimated from the software based on their pharmacokinetic data”, could you provide the data on the curve fittings and parameter estimations and used observed data in a supplementary information? This way, the uncertainty in the curve fittings could be evaluated.
Response:
Thanks for your advice. we have added the the data on the curve fittings and parameter estimations and used observed data in a supplementary information as following.
Table S1. Calculation of SF
|
Drug |
Reference of Obs data |
SF |
Mean |
|
CsA |
[23] |
0.01 |
0.01 |
|
|
[24] |
0.01 |
|
|
Flu |
[25] |
23 |
21 |
|
|
[25] |
22 |
|
|
|
[26] |
20 |
|
|
Sim |
[27] |
25.15 |
25 |
|
|
[28] |
25.9 |
|
|
|
[29] |
25.28 |
|
|
Lov |
[30] |
2.11 |
4 |
|
|
[25] |
4.21 |
|
|
|
[31] |
6.08 |
|
The calculated SF of cyclosporine, fluvastatin, simvastatin and lovastatin in Table 3 were estimated from the software based on their pharmacokinetic data showed in Table S1.
Point 5: However, whilst Ki,OATP was changed according to observed, the other parameters were not treated the same way. This gives an unfair comparison to the other changed parameters. Either change all parameters with the observed variability and evaluate how much this might affect the AUC and Cmax. OR change all parameters with the same fold, to have a fair comparison.
Response:
Thanks for your advice. We have added the result of 2-fold change of Ki,OATP into Table 6 to make it more complete.
Point 6: To improve readers’ understanding, could the following sentence “was used to assess model performance in this part, as well as pharmacokinetics parameters, Cmax and AUC.” be changed to “Model performance was assessed using absolute average fold error (AAFE) around each data point, as well as the ratio between observed and predicted Cmax and AUC”?
Response:
Thanks for your advice. We have changed our expression as you suggested. It’s really clearer than before.
“Model performance was assessed using absolute average fold error (AAFE) around each data point, as well as the ratio between observed and predicted Cmax and AUC.”
Point 7: Is it possible to add the rest of the Pred/obs figures as a supplementary materials, if it makes the figures in the result section to “messy”?
Response:
Thanks for your advice. We have added the the rest (atorvastatin Report 2 and lovastatin Report 2) of the Pred/obs in Supplementary Material Figure S1. The other Reports without the detail observed data in literature.
“The Pred/obs figures for atorvastatin Report 2 and lovastatin Report 2 are showing in Figure S1. While the others Reports without figures were the ones which have no observed plasma concentration curve.
Figure S1. Predicted plasma concentrations (line) of the victim drugs agents using PBPK model and observed plasma concentrations (points) following oral administration to subjects when coadministrated with CsA (twice daily). (A) atorvastatin; (B) lovastatin. Observed data were cited from clinic reports. [13][14]”
Point 8: It could this data be added to the manuscript, so the readers understand why this is the correct contribution order? Otherwise, the discussion is based upon results not shown.
Response:
Thanks for your advice. we have added the result caused by the Ki,OATP with two-fold change in Table 6. This may make it clearer.
Special thanks to you for your good comments again!
Reviewer 4 Report
Dear authors,
I highly appreciate your decision about reviewing the manuscript.
There are still several grammar corrections that should be revised in the manuscript as for example the following sentences. Nonetheless, I highly advice the authors to consider a thorough revision of the grammar.
"Hepatic uptake of these drugs is mainly mediated by hepatic OATPs…"
"The highlight of our work..."
"The predictions were compared with clinic reports (should be clinical)
"Atorvastatin, substrate of…"
"Special version for simvastatin,.."
"While the active metabolite simvastatin is..."
"The lag time of observations can be interpreted as the time for simvastatin (lactone) to simvastatin acid". Do the authors mean conversion? There is a missing verb in this sentence.
"The Pharmacokinetic profiles for coadministration of perpetrator CsA and victim drugs (atorvastatin, cerivastatin, pravastatin, rosuvastatin, fluvastatin, simvastatin, lovastatin, repaglinide and bosentan) were showed in Figure 2 and Table 5." Are shown?
"The criteria for to select victim drugs: 1. the victims selected should be as a drug in clinical; 2. all of which are substances of OATP and some of which are substance of CYP3A; 3. all of which have the clinical data with cyclosporine and some of which have strong DDI effect with cyclosporine A. Only the one that meet all conditions were selected."
"Fig1. Different points represented different dosages of drugs as showing in figure legend." The dots do not represent a dosage.
I find extremely long the description between 2.3.1 to 2.3.8 as all this information is already provided in a new table the authors incorporated to the revised version of the manuscript. Thus, I highly suggest to only leaving as a text, specific comments about this table.
Page 6, the legend figure has been removed and now it lacks of a legend. There are several flaws in this graph including the lack of fit of the models to the data corresponding to panels A, F, and G. Panel I has no observed data
Author Response
Dear Reviewer:
Thanks for your comments, which are all valuable and very helpful for revising and improving our paper, as well as the important guiding significance to our researches. We have studied comments carefully and have made correction which we hope meet with approval. Revised portion are marked in red in the paper. The main corrections in the paper and the responds to the reviewer’s comments are as flowing:
Point 1: "The lag time of observations can be interpreted as the time for simvastatin (lactone) to simvastatin acid". Do the authors mean conversion? There is a missing verb in this sentence.
Response:
Thanks for your advice. We have changed this sentence as following:
"The lag time of observations can be interpreted as the time for simvastatin (lactone) converse to simvastatin acid"
Point 2: "The Pharmacokinetic profiles for coadministration of perpetrator CsA and victim drugs (atorvastatin, cerivastatin, pravastatin, rosuvastatin, fluvastatin, simvastatin, lovastatin, repaglinide and bosentan) were showed in Figure 2 and Table 5." Are shown?
Response:
Thanks for your advice. We have changed this sentence as following:
"The Pharmacokinetic profiles for coadministration of perpetrator CsA and victim drugs (atorvastatin, cerivastatin, pravastatin, rosuvastatin, fluvastatin, simvastatin, lovastatin, repaglinide and bosentan) are showed in Figure 2 and Table 5."
Point 3: "The criteria for to select victim drugs: 1. the victims selected should be as a drug in clinical; 2. all of which are substances of OATP and some of which are substance of CYP3A; 3. all of which have the clinical data with cyclosporine and some of which have strong DDI effect with cyclosporine A. Only the one that meet all conditions were selected."
Response:
Thanks for your advice. We have changed this sentence as following:
"The criteria for to select victim drugs: 1. the victims drugs selected should be used in clinical at present; 2. all of which are substances of OATP and some of which are substances of CYP3A; 3. all of which have the clinical data with cyclosporine and some of which have strong DDI effect with cyclosporine A. Only the one that meet all conditions was selected."
Point 4: "Fig1. Different points represented different dosages of drugs as showing in figure legend." The dots do not represent a dosage.
Response:
Thanks for your advice. we are sorry for our incorrect writing which confuse you. We have changed this sentence as following:
"Different points represented different dosages for drugs in the corresponding figure "
Point 5: I find extremely long the description between 2.3.1 to 2.3.8 as all this information is already provided in a new table the authors incorporated to the revised version of the manuscript. Thus, I highly suggest to only leaving as a text, specific comments about this table.
Response:
Thanks for your advice. we have We've condensed the Result as following.
“2.3.1 DDI of CsA with atorvastatin
Atorvastatin is a substrate of OATPs, CYP3A4, BCRP, P-gp and MRP2. Renal efflux transporters (BCRP, P-gp and MRP2) also mediated renal excretion of atorvastatin, although its contribution to total clearance was minor. Coadministration of CsA induced DDI with atorvastatin via inhibiting CYP3A4 and above transporters. Two clinical studies for DDIs of CsA and atorvastatin were included in the study.
Report 1 [32,33]. Eighteen renal transplant patients received CsA-based immunosuppressive therapy. The CsA dosage was 5.20 (2.03–12.66) mg/kg/d. The patients orally received daily 10 mg atorvastatin for 4 weeks. Pharmacokinetics of atorvastatin was performed following 4-week treatment. Eight age-matched healthy subjects were selected for controls and administrated daily 10 mg atorvastatin for 1-week [32]. The results showed that simulated pharmacokinetic profile of atorvastatin following oral multidose of 40 mg atorvastatin was comparable to observed data (Figure. 2A). Predicted and observed steady state AUC0-24 and Cmax of atorvastatin in healthy subjects and in renal transplant patients were showing in Table 5. Dosages of CsA were set to be 886 mg/70 kg/d. The predictions were within the fold error range compared with clinical observations. The predicted AUCR and CmaxR values were in line with observations.
Report 2 [34]. Thirteen heathy subjects orally received 40 mg atorvastatin for 6 days. On evening of day 6, first dose (2.5 mg/kg) of CsA was administrated. On day 7, CsA (2.5 mg/kg) was co-administrated with atorvastatin at morning. Pharmacokinetics of atorvastatin after morning dose was implemented on day 6 and day 7. CsA dosage was set to be 175 mg/70 kg. The pharmacokinetic profiles and corresponding pharmacokinetic parameters of atorvastatin before and after second dose of CsA were predicted using the developed PBPK model. The results showed that simulated pharmacokinetic profile of atorvastatin following oral multidose of 40 mg atorvastatin was comparable to observed data. Predicted steady-state AUC0-6 and Cmax following multidose of atorvastatin to healthy subjects were within 0.5-2.0 folds of clinic observations (data shown in Table 5). Coadministration of CsA increased plasma exposure to atorvastatin. The predicted AUC0-6 and Cmax of atorvastatin following co-administration of CsA were lower than the clinic observations. The predicted AUCR and CmaxR were also lower than observed data.
2.3.2 DDIs of CsA with cerivastatin
Cerivastatin is a substrate of P-gp, BCRP, OATP1B1 and CYP2C8. CsA-mediated DDI with cerivastatin is attributed to inhibiting OATP1B1-mediated hepatic uptake and P-gp/BCRP-mediated intestinal efflux. A clinic report [35] showed that 12 renal transplant recipients received CsA-based immunosuppressive therapy. Dosage regimen of CsA ranged from 75 mg to 225 mg twice daily. The patients were administrated 0.2 mg cerivastatin daily for 7 days. Pharmacokinetics of cerivastatin was implement on day 1 and day 7. Twelve healthy subjects were administrated single dose of cerivastatin, serving as the control group. The predicted Cmax and AUC of cerivastatin following oral 0.2 mg of cerivastatin were good agreement with the measured data in healthy subjects. Dosage of CsA was assumed 225 mg/d. The predicted Cmax and AUC of cerivastatin following oral 0.2 mg of cerivastatin to renal transplant recipients on day 1 and day 7 were fell within 0.5-2.0 folds of observations, shown in Figure 2B and Table 5. Compared with healthy subjects, the predicted AUCR on day 7 was 5.2, falling within 0.5-2.0 folds of observation (4.8), but predicted CmaxR was 2.3, less than 0.5-fold observations (5.0).
2.3.3 DDIs of CsA with pravastatin
Pravastatin is a substrate of MRP2 and OATPs. The drug mainly eliminates via MRP2-mediated biliary and MRP2-mediated renal secretion, indicating that CsA leads to DDI with pravastatin via affecting OATP-mediated hepatic uptake, MRP2-mediated intestinal efflux, MRP2-mediated biliary excretion and MRP2-mediated renal secretion. Two studies for DDIs of CsA with pravastatin were included in the study.
Report 1 [36]. Twenty-three stable kidney allograft recipients undergoing CsA-based immunosuppressive therapy received a single 20 mg oral daily dose of pravastatin for 28 days. Pharmacokinetics of pravastatin were implemented on day 1 and day 28 following pravastatin administration. CsA was administered as oral capsule formulation twice daily, whose dosage was set to be 420 mg/d. The pharmacokinetic profiles of pravastatin following multidose of 20 mg to patients on day 1 and day 28 were predicted and compared with observations (Figure. 2C). The Cmax and AUC0-24 of pravastatin co-administrated with CsA were predicted in Table 5. The predictions were fell within 0.5-2.0 folds of clinic observations. The mean values in Table 4 were normalized by 20 mg, as controls, AUCR and CmaxR on day 28 were calculated to 3.1 and 3.6, respectively, within the fold error range compared with predictions (3.7 and 4.1 respectively).
Report 2 [37]. Eleven heart-transplant recipients underwent CsA-based immunosuppressive therapy. The patients received a daily dose of 40 mg/d pravastatin for the first 8 days, then was reduced to 10 mg/d administered until day 29. Pharmacokinetics of pravastatin was carried out on day 1, 8 and 29. Pharmacokinetics of pravastatin in 8 healthy subjects following a single dose of 60 mg pravastatin was also investigated. Dosage of CsA in patients were set to 400 mg/d. The predicted Cmax and AUC0-24 values normalized by 10 mg pravastatin in healthy subjects were in line with clinic observations. The predicted 10 mg-normalized Cmax values of pravastatin co-administrated CsA on days 1, 8 and 29 were also within the fold error range compared with observations. The predicted 10 mg-normalized AUC0-24 values in heart-transplant on day 1, 8 and 29 were lower than observations. The predicted CmaxR were falling within 0.5-2.0 folds observations but estimated AUCRs were lower than observations.
2.3.4 DDI of CsA with rosuvastatin
Rosuvastatin is a substrate of P-gp, BCRP, MRP2 and OATPs. Rosuvastatin mainly eliminates via BCRP-dependent biliary and renal secretion. Thus, DDI of CsA with rosuvastatin may be due to inhibition of these transporter function. A report [38] demonstrated that heart transplanted patients undergoing a standard scheme of CsA-based therapy and 10 healthy subjects received single dose of 10 mg rosuvastatin, followed by once-daily oral dose of 10 mg rosuvastatin for 10 days. Another group might receive dose of 20 mg rosuvastatin, followed by once-daily oral dose of 20 mg rosuvastatin for 10 days. Pharmacokinetics of rosuvastatin was documented at single dose and last dose for patients and healthy subjects. CsA dosage was set to be 200 mg twice a day. The pharmacokinetics of rosuvastatin in healthy subjects and patients were predicted and compared with observation (Figure 2D and Table 5). The predicted Cmax and AUC of rosuvastatin following multidose 10 mg rosuvastatin to healthy subjects were agreed with observations. The predicted Cmax and AUC of single dose and multidose of 10 mg to patients were lower than that for measured geometric mean. As same as single dose and multidose of 20 mg rosuvastatin to patients. The predicted AUCR and CmaxR following multi-doses to patients were 2.4 and 2.9, respectively, which were also lower than observations.
2.3.5 DDIs of CSA with fluvastatin
Fluvastatin is metabolized by CYP2C9. Fluvastatin is also substrate of P-gp, BCRP, MRP2 and OATPs. Liver BCRP also mediates biliary excretion of fluvastatin. Thus DDI of CsA with fluvastatin mainly results from inhibiting above transporters. Two clinic reports were cited.
Report 1 [39]. Ten heart transplant patients under CsA-based therapy and 10 healthy subjects received 40 mg fluvastatin daily for 28 days. Pharmacokinetics was documented on day 1 and day 28. CsA dosage was assumed to be 200 mg/d. The plasma concentrations of fluvastatin in patients and healthy subjects were predicted and compared with observations (Figure 2E and Table 5). The predicted Cmax and AUC0-24 on day 1 and day 28 following oral 40 mg fluvastatin to healthy subjects were within the fold error range compared with the observations. The predicted Cmax and AUC0-24 on day 1 and day 28 following oral 40 mg of fluvastatin to heart transplant patients were also fell within clinical observations. On day 28, predicted CmaxR and AUCR were 3.4 and 3.2, respectively, which were in line with observations (6.0 and 3.1, respectively).
Report 2 [40]. Twenty hypercholesterolemic renal transplant recipients receiving CsA in combination with azathioprine and methylprednisolone administrated 20 mg fluvastatin daily for 4-6 weeks. CsA dosage was set to 200 mg/d. The predicted Cmax value was falling within 0.5-2.0 folds of observation, but predicted AUC was 2 times higher than observation.
2.3.6 DDI of CsA with simvastatin
Simvastatin is a substrate of OATPs and CYP3A4, which mainly eliminates via CYP3A4-mediated metabolism mechanism. CsA leads to DDI with simvastatin via inhibiting OATP-mediated hepatic uptake, CYP3A4-mediated metabolism in intestine and liver. Two clinic reports were included in the simulation.
Report 1 [41]. Pharmacokinetics of simvastatin following oral dose 20 mg was implemented in five transplant patients receiving CsA (1.1~3.8 mg/kg) and 5 patients without CsA. Dosage of CsA was set to be 266 mg/70 kg. The pharmacokinetic profiles of simvastatin following oral 20 mg simvastatin to patients treated with CsA or without CsA were predicted and compared with observations (Figure 2F and Table 5). The predicted AUC0-24 and Cmax values in patients treated with CsA were fell within ranges of observed AUC0-24 and Cmax. The predicted AUC0-24 and Cmax values in patients treated without CsA were lower than clinic observations. Predicted AUCR and CmaxR were 3.6 and 7.3, respectively, which were higher than observations (2.7 and 2.08). However, using observed mean values normalized by 20 mg in Table 4, AUCR values of Cmax and AUC were calculated to 6.5 and 6.9, respectively, within the fold error range compared with predictions (7.3 and 3.6. respectively).
Report 2 [42]. Seven hypercholesterolemic heart transplant patients under CsA-based immunosuppressive therapy and 7 hypercholesterolemic non-heart transplant patients (served as control) were included. Both groups undergoing 10 mg simvastatin treatment for 6 weeks, attended the clinic in the morning after an overnight fast and received a double dose (20 mg). Dosage of CsA was set to be 200 mg daily twice. The predicted concentrations of simvastatin acid at 1, 2 and 3 h following oral 20 mg of simvastatin to heart transplant patients were higher than those in non-heart transplants. The observed mean plasma concentrations of simvastatin acid in heart transplant patients at 1, 2, and 3 h following oral 20 mg simvastatin was higher than those in non-heart transplants. The plasma concentrations of simvastatin acid in non-heart transplants were lower than 2 ng/mL. The predictions were higher than the observations.
2.3.7 DDI of CsA with lovastatin
Lovastatin is a substrate of OATPs, MRP2 and CYP3A4, which mainly eliminates via CYP3A4-mediated metabolism mechanism. DDI of CsA with lovastatin occurs via inhibiting OATP-mediated hepatic uptake, MRP2-mediated intestinal efflux and CYP3A4-mediated metabolism. Three clinic reports were cited in the study.
Report 1 [36]. Twenty-one stable kidney allograft recipients undergoing CsA-based immunosuppressive therapy received a single 20 mg oral daily dose of lovastatin for 28 days. CsA dosage was between 2 and 6 mg/kg/day. Pharmacokinetic profiles of lovastatin were implemented on day 1 and day 28. CsA was administered as oral capsule formulation twice daily. Pharmacokinetic profiles of lovastatin co-administrated 420 mg CsA were simulated and compared with observations (Figure 2G and Table 5). The predicted Cmax and AUC of lovastatin on day 1 and day 28 were lower than the observed medium values. Using data list in Table 4 as controls, the calculated CmaxR and AUCR were within the fold error range compared with predicted.
Report 2 [43]. Five types of patients were investigated as follows: (1) 6 heart transplant recipients receiving CsA (273 mg)-based triple immunosuppressive regimen; (2) five kidney transplant recipients receiving CsA (230 mg)-based triple immunosuppressive regimen; (3) fiver patients with psoriasis treated with CsA (290 mg) monotherapy; (4) five kidney transplant recipients receiving prednisolone and azathioprine; and (5) eight hypercholesterolemic patients serving as “control”. These patients orally received 10 mg lovastatin once daily for 10 days. Pharmacokinetics of lovastatin was implemented on day 10. The simulated AUC0-8 values in heart transplant recipients receiving CsA, kidney transplant recipients receiving CsA, patients with psoriasis treated with CsA, and hypercholesterolemic patients were 31.33, 27.87, 32.64, and 3.89 ng‧h/mL. The measured mean AUC0-8 values in heart transplant recipients receiving CsA, kidney transplant recipients receiving CsA, patients with psoriasis treated with CsA and hypercholesterolemic patients were 175, 110, 110 and 26 ng‧h/mL. Compared with hypercholesterolemic patients, AUCRs of lovastatin following co-administration of CsA were calculated to be 6.7, 4.2 and 4.2, respectively. The predicated AUCRs of lovastatin in heart and kidney transplant recipients and patients with psoriasis were 8.1, 7.2 and 8.4, respectively. All of predictions were fell within 0.5-2.0 folds of observations.
Report 3 [44]. Twenty-four (14 cardiac and 10 renal) transplanted patients under CsA based-triple immunosuppressive regimen, received lovastatin treatment. Dosage of CsA was 300 (150-400) mg/d for heart transplanted patients and 255 (200-400) mg/d for renal transplanted patients. Five milligrams lovastatin was then given once daily in the morning for 3 weeks. The dose was thereafter increased by 5 mg every third week until 20 mg once daily, which was continued until week 18. Blood levels of lovastatin were measured prior to and 2 h after oral CsA and the first dose of lovastatin at each dose level. Blood levels of lovastatin in hypercholesterolaemic non-transplanted patients prior to and 2 h after 10 mg lovastatin were served as control. Dosages of CsA was set to be 300 mg. The predicted concentrations of lovastatin at 2 h following lovastatin 5 mg at weeks 0 and 3, 10 mg at week 6, 15 mg at week 9, and 20 mg at weeks 12 and 18) were 5.09, 5.93, 12.27,18.91, 24.77 and 25.70 ng/mL, respectively. They were within 0.5-2.0-fold of observations. The predicted concentrations at 2 h following lovastatin 10 mg to hypercholesterolaemic non-transplanted patients was 1 ng/mL, lower than the observation concentration 4.6 ng/mL. Predicted AUCR using concentration at 2 h following 10 mg to transplanted patients at 6 weeks was 12, which was higher than observation (4.6).
2.3.8 DDI of CsA with repaglinide
Repaglinide eliminates mainly via CYP2C8. To some extents, CYP3A4 also mediates repaglinide metabolism. Repaglinide is also a substrate of OATP1B1 and P-gp, indicating that DDI of CsA with repaglinide is involved in inhibition of CYP3A4, P-gp, and OATPs.
A report [45] showed that 12 healthy subjects at PM 8 on day 1 and at 8 AM on day 2, received 100 mg CsA or placebo. At 9 AM on day 2, they ingested 0.25 mg dose of repaglinide. Pharmacokinetics of repaglinide was documented. The plasma concentrations of repaglinide following 0.25 mg dose of repaglinide alone or coadministration of CsA were simulated and compared with observations (Figure 2 H and Table 5). The predicted Cmax and AUC of repaglinide when co-administrated CsA were 2.4- and 2.5-fold higher than those following repaglinide alone. The measured Cmax and AUC of repaglinide when co-administrated CsA were 1.7 folds and 2.4 folds higher than those following repaglinide alone. All predictions fell within 0.5-2.0 folds of observations.
2.3.9 DDI of CsA with bosentan
Bosentan mainly eliminates via metabolism mechanism, which partly is attributed to CYP3A4. Bosentan is also a substrate of MRP2 and OATPs and renal MRP2 may be involved in its renal excretion of bosentan. These results inferred that DDI of CsA with bosentan is attributed to inhibition of CYP3A4-mediated metabolism, OATPs-mediated hepatic uptake and MRP2-mediated renal secretion.
A report [46] demonstrated that bosentan was given to 10 healthy subjects twice daily at the dose of 500 mg for 8 days. The subjects received 300 mg CsA twice daily starting from the day 1 evening for 8 days. On days 1 and 8, pharmacokinetics of bosentan was documented following the morning dose. The plasma concentrations of bosentan were stimulated (Figure 2I and Table 5). The predicted Cmax and AUC of bosentan following coadministration of CsA were 2.2-fold and 3.76-fold higher than those before co-administration of CsA. The measured Cmax and AUC of bosentan following coadministration of CsA were higher than thosebefore coadministration of CsA. Predicted CmaxR and AUCR were within the fold error range compared with the observed values.”
Point 6: Page 6, the legend figure has been removed and now it lacks of a legend. There are several flaws in this graph including the lack of fit of the models to the data corresponding to panels A, F, and G. Panel I has no observed data
Response:
Thanks for your advice. we have added the legend in Figure 2. The observed data of bosentan with CsA was not shown in literature, so we haven’t got the observed data point.
Special thanks to you for your good comments again!